# Coherent optical coupling to surface acoustic wave devices

Arjun Iyer ●[1] ✉, Yadav P. Kandel ●[2], Wendao Xu ●[1], John M. Nichol ●[2] & William H. Renninger[1,2]

Surface acoustic waves (SAW) and associated devices are ideal for sensing, metrology, and hybrid quantum devices. While the advances demonstrated to date are largely based on electromechanical coupling, a robust and customizable coherent optical coupling would unlock mature and powerful cavity optomechanical control techniques and an efficient optical pathway for long-distance quantum links. Here we demonstrate direct and robust coherent optical coupling to Gaussian surface acoustic wave cavities with small mode volumes and high quality factors ($>10^5$ measured here) through a Brillouin-like optomechanical interaction. High-frequency SAW cavities designed with curved metallic acoustic reflectors deposited on crystalline substrates are efficiently optically accessed along piezo-active directions, as well as non-piezo-active (electromechanically inaccessible) directions. The precise optical technique uniquely enables controlled analysis of dissipation mechanisms as well as detailed transverse spatial mode spectroscopy. These advantages combined with simple fabrication, large power handling, and strong coupling to quantum systems make SAW optomechanical platforms particularly attractive for sensing, material science, and hybrid quantum systems.

Surface acoustic wave devices based on electrically driven piezo-electric materials are essential to modern technologies, including for communications[1–3] and chemical and biological sensors[4,5]. SAWs have more recently emerged as an exciting resource for quantum systems[6–9] because of their low loss, tight surface confinement, and strong coupling to a variety of quantum systems. As universal quantum transducers[6,7], SAWs and associated manipulation and probing techniques have been demonstrated in color centers[10], superconducting qubits[11–14], semiconductor quantum dots[15–17], 2D materials[18–20], and superfluids[21,22]. While electrical control of SAWs has matured over the past several decades[1], robust and customizable coherent optical coupling has not yet been demonstrated. Coherent optical coupling would enable powerful techniques established in cavity optomechanical systems, such as quantum transduction[23,24], quantum-limited force and displacement sensing[25–29], generation of non-classical states of optical and acoustic fields[30–32] and ground state cooling of mechanical resonators[33–36]. In addition, robust optomechanical coupling to SAW

devices would enable an ideal optical pathway for long-distance quantum links, a longstanding goal of experimental quantum information science[37–39].

While optical surface Brillouin scattering has been successful in probing incoherent thermal surface phonons for the study of thin films[40,41], many optomechanical applications, including for quantum science, require coherent and stimulated interactions. Brillouin processes have recently enabled coherent coupling to bulk acoustic modes with record lifetimes in shaped crystals, but with the low optomechanical coupling associated with larger bulk mode volumes[42,43]. In nanoscale systems designed for cavity optomechanics, large coupling strengths are available, but often with complex designs, including sub-wavelength and suspended structures which can be challenging to integrate into larger hybrid quantum devices. In addition, nanoscale confinement also leads to undesirable heating effects[9,43,44], which limit photon numbers and acoustic quality factors, and can require complex optical pumping schemes[45,46]. SAW devices

[1]Institute of Optics, University of Rochester, Rochester, NY, USA. [2]Departament of Physics and Astronomy, University of Rochester, Rochester, NY, USA. ✉e-mail: aiyer2@ur.rochester.edu

combine intrinsically tight confinement on the surface of bulk substrates with the potential for high-power handling and simple fabrication. In recent SAW-based microwave-to-optical transduction schemes[9,23,47,48], electrically generated SAWs are coupled to acoustic modes of a distinct resonator, such as a nanomechanical phononic crystal cavity. While demonstrating exceptional optomechanical coupling strengths, these devices achieve low net efficiencies, primarily limited by the phonon injection efficiency of the electrically generated SAWs to the acoustic cavity modes[7,49,50]. In an alternative approach, Okada et al.[51] examined cavity optomechanical systems mediated directly with SAWs. However, without optomechanical phase matching, efficient SAW confinement, and modal size matching between optical and acoustic fields, the system is limited to lower phonon frequencies, mechanical quality factors, and coupling strengths. Optomechanical coupling to SAW whispering gallery modes in microresonators[52,53], travelling-wave SAWs within tapered-optical fibers[54], and SAWs driven by optical absorption and thermal relaxation of metallic electrodes[55,56] also enable several new possibilities. However, these devices will be challenging to integrate with a range of qubit systems, which require specific geometries and minimal heating. Direct, efficient, coherent optical access to simple high-quality and high-power handling SAW devices will be an important step toward realizing the full promise of classical and quantum SAW-based technologies.

Here, we establish a frequency-tunable Brillouin-like optomechanical coupling with integrable surface acoustic wave devices that is direct, coherent, power-tolerant, and efficient. The technique is demonstrated with simple single-crystalline substrates supporting long-lived Gaussian SAW cavity modes confined with deposited curved metallic grating mirrors. Strong optomechanical coupling is demonstrated by engineering phase-matched Brillouin-like interactions between the trapped acoustic modes and incident out-of-plane non-collinear optical fields. In contrast to previous studies on SAW resonators, the demonstrated technique does not require piezoelectricity and can be applied to practically any crystalline media, which we demonstrate by optically driving piezo-inactive SAW devices. This approach, therefore, enables access to high-Q surface acoustic modes on quantum-critical materials that are not piezo-active, such as diamond and silicon. In addition, the absence of interdigital transducers or any other acousto-activating device enables cavity-limited quality-factors within small mode volume cavities, including the record ~120,000 quality-factors demonstrated in GaAs at cryogenic temperatures in this report. The presented cavities, operating at 500 MHz, can be tuned through several GHz by varying the incident angle of the optical beams through simple phase matching (momentum conservation) considerations. The frequency and material versatility of this coupling technique is well suited for materials spectroscopy, as illustrated here through measuring and characterizing phonon dissipation mechanisms in GaAs cavities with metallic mirrors. The SAW optomechanical platform presented combines the simplicity and power-handling advantages of bulk optomechanical systems with the small acoustic mode volumes of nanoscale systems for enhanced interaction strengths enabling a multi-functional integrated platform for sensing, quantum processing, and condensed matter physics.

## Results

### Non-collinear Brillouin-like coupling to surface acoustic waves

The SAW device consists of a Fabry-Perot Gaussian surface acoustic wave cavity on a single-crystalline substrate formed by two acoustic mirrors composed of regularly spaced curved metallic reflectors. Two non-collinear optical beams, a pump field, and a Stokes field, are incident in the region enclosed by the acoustic mirrors (Fig. 1a). The confined Gaussian surface acoustic mode can mediate energy transfer between the two optical fields provided phase-matching (momentum conservation) and energy conservation relations are satisfied, as is the

case with Brillouin scattering from bulk acoustic waves[57]. For pump and Stokes fields with wavevector (frequency) $\vec{k}_p(\omega_p)$ and $\vec{k}_s(\omega_s)$, respectively, that subtend equal but opposite angles, θ, with respect to the surface normal (z-axis), the optical wavevector difference can be approximated as $\Delta\vec{k} \approx 2k_0 \sin\theta\hat{x}$, assuming $k_p \approx k_s = k_0$ and for $\hat{x}$, a unit vector parallel to the surface; the corresponding optical frequency difference is $\Delta\omega = \omega_p - \omega_s$ (Fig. 1b). Note that the magnitude of the optical wavevector difference is tunable by the optical angle of incidence. For the case of freely propagating surface acoustic waves, the acoustic dispersion relation is linear and can be expressed as $\Omega = qv_R$, where Ω,q, and $v_R$ are the phonon frequency, phonon wavevector, and Rayleigh SAW velocity, respectively. The phase-matched phonon wavevector ($q_0$) and frequency ($\Omega_0$) are then given by the relations: $q_0 = \Delta k = 2k_0 \sin\theta$ and $\Omega_0 = q_0 v_R$. A propagating SAW, therefore, yields a single-frequency optomechanical response, similar to the standard Brillouin response in bulk materials from propagating longitudinal waves. However, the accessible phonon spectrum is significantly modified in the presence of a surface acoustic cavity and optical beams with finite beam sizes, as illustrated by the modified acoustic dispersion plot in Fig. 1c. First, because standing SAW cavity modes are formed, the phonon wavevectors and frequencies become discretized to specific values $q_m = \frac{m\pi}{L_{eff}}$ and $\Omega_m = q_m v_R$, respectively, characterized by mode number $m$, where the free spectral range of the cavity is $\Delta\Omega = \frac{\pi v_R}{L_{eff}}$, and $L_{eff}$ is the effective cavity length. Second, unlike ideal mirrors, acoustic Bragg mirrors only efficiently confine a finite number of longitudinal modes (blue circles in Fig. 1c) determined by the reflectance and periodicity of the metallic reflectors[i]. Finally, because the optical fields are Gaussian beams with finite spatial extents, appreciable optomechanical coupling exists over a range of optical wavevectors values centered around the phase-matched configuration, $\Delta k = q_m$. The effective optomechanical coupling rate to the cavity mode $m$, $g_0$, varies as a function of optical wavevector mismatch as $g_0(\Delta k) \propto \exp(-(\Delta k - q_m)^2/\delta k^2)$, where $\delta k = 2\sqrt{2}/r_0$ and $r_0$ is the radius of incident optical fields. Equivalently, the coupling rate can be expressed as a function of the angle of incidence as $g_0(\theta) \propto \exp(-(\theta - \theta_m)^2/\delta\theta^2)$, for small angles such that $\sin\theta \approx \theta$ and where $\theta_m$ is the phase-matching angle of the acoustic cavity mode given by $\theta_m = \frac{q_m}{2k_0}$. The corresponding angular bandwidth is given as $\delta\theta = \frac{\delta k}{k_0} = \frac{\sqrt{2}}{r_0 k_0}$ (see section S2 of Supplementary Information). The resultant optomechanical spectrum consists of several discrete resonances from SAW cavity modes which lie both within the acoustic mirror and optical phase-matching bandwidths (unconfined radiative longitudinal modes are indicated by gray circles in Fig. 1c).

Gaussian SAW cavities are designed to achieve small acoustic mode volume and appreciable coupling strengths (see "Methods" section and section S3 of Supplementary Information). Diffraction losses are mitigated by accounting for the anisotropy of the acoustic group velocity on the underlying crystalline substrate[58]. GaAs is chosen because of its large photoelastic response, ease of fabrication, and integration with other quantum systems such as qubits. 3-dimensional numerical finite element simulations are performed of a SAW cavity on [100]-cut GaAs oriented along [100]-direction. The acoustic wavelength of $\lambda_a = 5.7\,\mu m$ and Gaussian waist, $w_a = 3\lambda_a$, are near-identical to the experimental devices described below, while the number of reflectors and the mirror spacing are reduced to maintain computational feasibility (see section S3 of Supplementary Information). The simulated cavities display a series of stable SAW cavity modes with Hermite-Gaussian-like transverse profiles (Fig. 1d–f) separated by the free spectral range of the cavity. As expected, the modes are confined to the surface and steeply decay into the bulk of the substrate (e.g., lower panel Fig. 1d). The observed beam waist of the fundamental Gaussian mode agrees well with the designed full-waist ($2w_a$) of $6\lambda_a$.

Higher-order anti-symmetric (Fig. 1e) and symmetric (Fig. 1f) mode solutions are also observed.

## Optomechanical spectroscopy of SAW cavities

To demonstrate coherent optical coupling to SAW devices, Gaussian SAW cavities are fabricated (see "Methods" section and section S4 of Supplementary Information) on a single crystal GaAs substrate (inset Fig. 2a). Optomechanical measurements are made at a temperature of $T = 4$ K for two sets of cavities, one oriented along the crystalline [110] direction, which is piezo-active, and one along the [100] direction, which is piezo-inactive. The cavities are designed for an acoustic wavelength of $\lambda_a \approx 5.7\,\mu$m ($\theta \approx 7.8°$), acoustic waist of $w_a \approx 4\lambda_a$ and mirror spacing of $L \sim 500\,\mu$m. The cavity parameters are chosen to optimize for practical constraints including finite optical apertures, electronics bandwidths, and the optical beam sizes. The effective cavity length ($L_{eff}$) is calculated to be $\sim 620\,\mu$m by accounting for the penetration depth into the mirrors[1,59]. The large mirror separation relative to the optical beam size minimizes absorptive effects arising from spatial overlap of optical fields with acoustic metallic reflectors (see section S9 of Supplementary Information).

The stimulated optomechanical response is measured with the sensitive phonon-mediated four-wave mixing measurement technique described in "Methods" section and Section S5 of Supplementary Information. The measured signal is a coherent sum of frequency-independent four-wave mixing in the bulk of the crystalline substrate and the optomechanical resonance, giving rise to Fano-shaped resonances[60]. The spectral response, at $T = 4$ K, of the piezo-inactive (active) cavity along the [100] ([110])-direction in Fig. 2a, b (c, d) reveals several equally spaced resonances over a wide spectral range centered at 480 MHz (506 MHz) separated by 2.3 MHz (2.4 MHz), which

corresponds to the free-spectral range ($v_R/2L_{eff}$) of the SAW cavity. The observed resonances span $\sim 10$ MHz, which is consistent with the designed acoustic mirror bandwidth. High-resolution spectral analysis of one of the observed SAW cavity resonances of the piezo-inactive (active) cavity (Fig. 2b (d)) reveals a spectral width, $\Gamma/2\pi$, of 4 kHz (76 kHz), corresponding to an acoustic quality factor of 120,000 (6700). The difference in resonance line shapes for the two cavity types could result from different phases relative to the coherent background (see section S12 of Supplementary Information). The measured traveling-wave zero-point coupling rate[42,61], $g_0$, for the piezo-inactive (active) cavity of $2\pi \times 1.4$ kHz ($2\pi \times 2.4$ kHz) is consistent with predicted values of $2\pi \times 1.7$ kHz ($2\pi \times 1.8$ kHz) obtained using known material parameters in conjunction with the device geometry (see sections S1 and S7 of Supplementary Information). Note that while $g_o$ is comparable between the two orientations, since the optical response is proportional to Q (see section S1 of Supplementary Information), the measured signal-to-noise ratio is better for [100]-oriented SAW devices with larger acoustic quality factors. This dependence of signal strength on quality factor also results in variation in signal strength for different axial modes (most notably in Fig. 2a) which have different confinement strengths depending on their spectral location relative to the acoustic mirror stop-band. As expected, no measurable acoustic response is observed when either of the optical drive tones is turned off. Additionally, as predicted by theoretical coupling calculations (see section S1 of Supplementary Information), no resonance is observed when the acoustic drives are orthogonally polarized to each other, or when the LO is orthogonally polarized to the incident probe (see section S11 of Supplementary Information). The demonstrated acoustic quality factors of the SAW devices are among the highest measured for focused SAW cavities on any substrate, corresponding to an $fQ$ product of

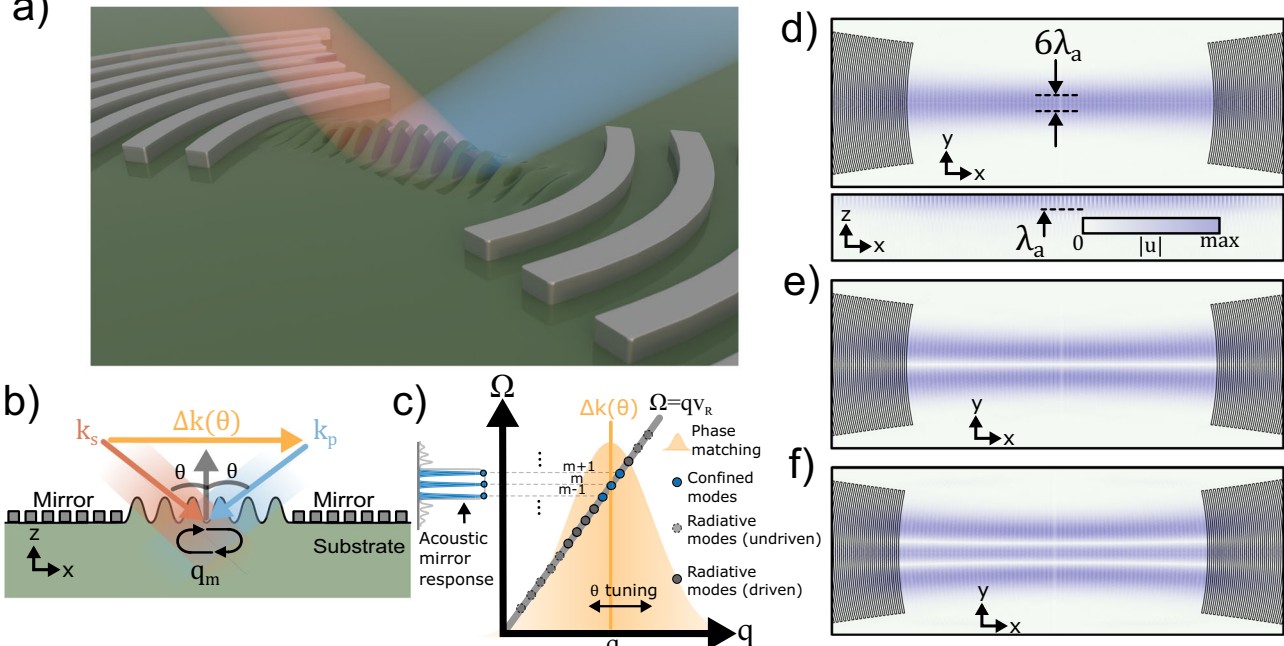

**Fig. 1 | Parametric optomechanical interactions mediated by Gaussian SAW resonators. a** Two non-collinear traveling optical fields are incident on a Fabry-Perot type Gaussian SAW resonator; interaction between the two optical fields is mediated by a Gaussian SAW cavity mode confined to the surface of the substrate. **b** Phase-matching diagram of the parametric process. The vectorial optical wave-vector difference, $\Delta\vec{k} = \vec{k}_p - \vec{k}_s = 2k_0 \sin\theta\,\hat{x}$, is angle-dependent and points along the direction of SAW cavity axis. **c** The acoustic dispersion relation $\Omega(q)$ is discretized in the presence of a SAW cavity. The final optomechanical response is determined by the modes, which are both within the phase-matching and the

acoustic mirror bandwidth (blue dots), while radiating longitudinal modes excluded by the acoustic mirror and the optical phase matching (gray dots) do not yield an optomechanical response. **d** Finite element calculation of the acoustic displacement magnitude, $|u|$, in a SAW cavity along [100] direction on [100]-cut GaAs illustrating the Gaussian mode (YX cross-Ssection ($z = 0$), upper panel) with the designed acoustic waist of $w_a = 3\lambda_a$ and an approximate penetration depth of $\sim \lambda_a$ (ZX cross-section ($y = 0$), lower panel). Panels **e** and **f** display YX ($z = 0$) cross-sections of acoustic displacement for (**e**) anti-symmetric and (**f**) symmetric higher-order transverse modes of the SAW cavity.

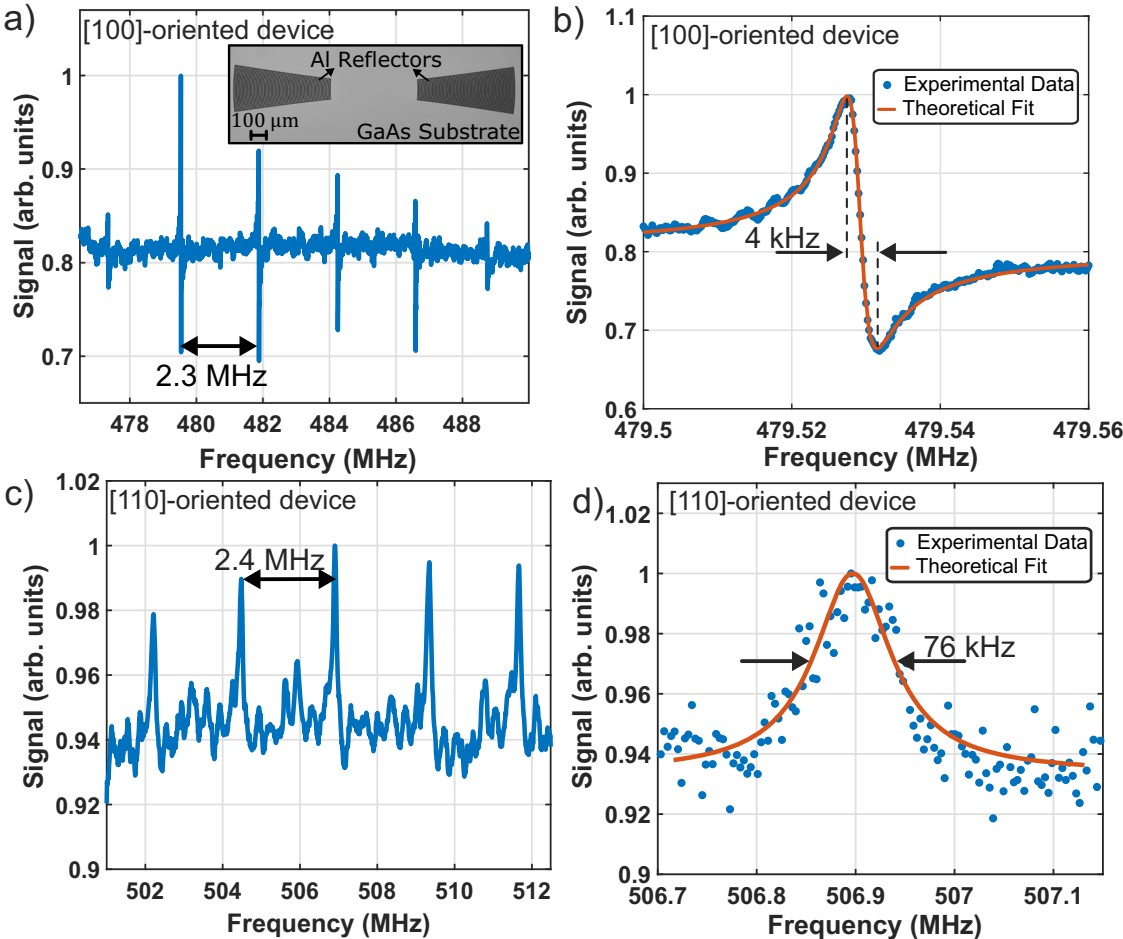

**Fig. 2 | Optically measured SAW devices.** Optomechanical response of 480 MHz SAW cavities oriented along the [100]-direction on [100]-cut GaAs at 4 K with (**a**) a wide frequency sweep revealing several discrete SAW cavity resonances separated by the 2.3 MHz cavity free spectral range (microscope image of the SAW device is inset), and (**b**) a high-resolution frequency sweep revealing an acoustic Q-factor of 120,000 or a spectral linewidth of ∼ 4 kHz. **c**, **d** Show similar measurements for the 505 MHz SAW cavity oriented along [110]-direction on [100]-cut GaAs. **c** A wide scan reveals SAW cavity modes separated by the free spectral range of 2.4 MHz and (**d**) the SAW modes on the [110]-oriented devices exhibit a maximum quality factor of 6700 with a corresponding linewidth of 76 kHz.

$6 \times 10^{13}$ Hz, which is also comparable to that of the best electro-mechanical focused Gaussian SAW devices[15,59,62]. Moreover, the accessed SAW cavity modes are along electromechanically inaccessible directions, demonstrating a key merit of the coherent optical coupling in enabling access to long-lived SAW modes regardless of their piezoelectric properties. The larger relative loss in the [110]-oriented cavities is consistent with excess ohmic loss from the metallic reflectors owing to non-uniform strain from the Gaussian modes and the resulting piezoelectric potential on the reflectors[63–65].

While the high-order spatial modes of a Gaussian SAW resonator are challenging to probe electromechanically, the coherent optomechanical technique allows for precise and direct excitation of spatial modes through fine control of the optical spatial overlap with specific acoustic mode profiles. Since transverse acoustic modes are spatially orthogonal, if the optical beams, and their resulting optical force distribution is mode-matched to the fundamental acoustic mode, it is also orthogonal to the higher-order acoustic modes. As a result, lateral displacement of the optical beams away from the cavity axis is necessary to observe optomechanical coupling to a specific higher-order SAW cavity mode (green in Fig. 3a, b), in addition to the response from the fundamental mode (red in Fig. 3a, b). The frequency separation between the fundamental and corresponding higher-order mode of 1.4 MHz is consistent with the predicted difference of 1.4 MHz. The exquisite spatial control available through optical techniques

could form the basis of novel SAW-based spatially resolved sensing and metrology.

Finally, the accessible phonon-mode bandwidth determined by phase matching is characterized through measurements of the Brillouin coupling coefficient ($G_B \propto |g_0|^2$) as a function of the angle of incidence of the optical fields (see section S2 of Supplementary Information). The coupling strength exhibits a Gaussian dependence on the angle (Fig. 3c) for peak coupling centered at $\theta_0 = 7.8°$, with an angular bandwidth of 0.9°, which agrees with the predicted bandwidth of 0.8°, obtained using experimental beam radius of $r_0 \approx 30\,\mu m$. The peak coupling at $\theta_0 = 7.8°$ is determined by the angle of incidence of the optical fields and the resultant wavevector difference, but the acoustic mode frequencies are independently fixed by the cavity geometry and the acoustic mirror response. The effective optomechanical coupling rate is maximized when the center of the optical phase matching envelope coincides with the peak reflection frequency of the acoustic mirrors (point outlined with purple circle in Fig. 3c and illustrated in the left panel of Fig. 3d) and decreases as they are mismatched (cyan circle in Fig. 3c and illustrated in the right panel of Fig. 3d). Because the optomechanical gain bandwidth and associated driven acoustic modes result from the optical spatial profiles, this technique presents the unique capability of tailoring the optomechanical gain profile for specific applications, from multi-mode optomechanics to tunable single frequency applications.

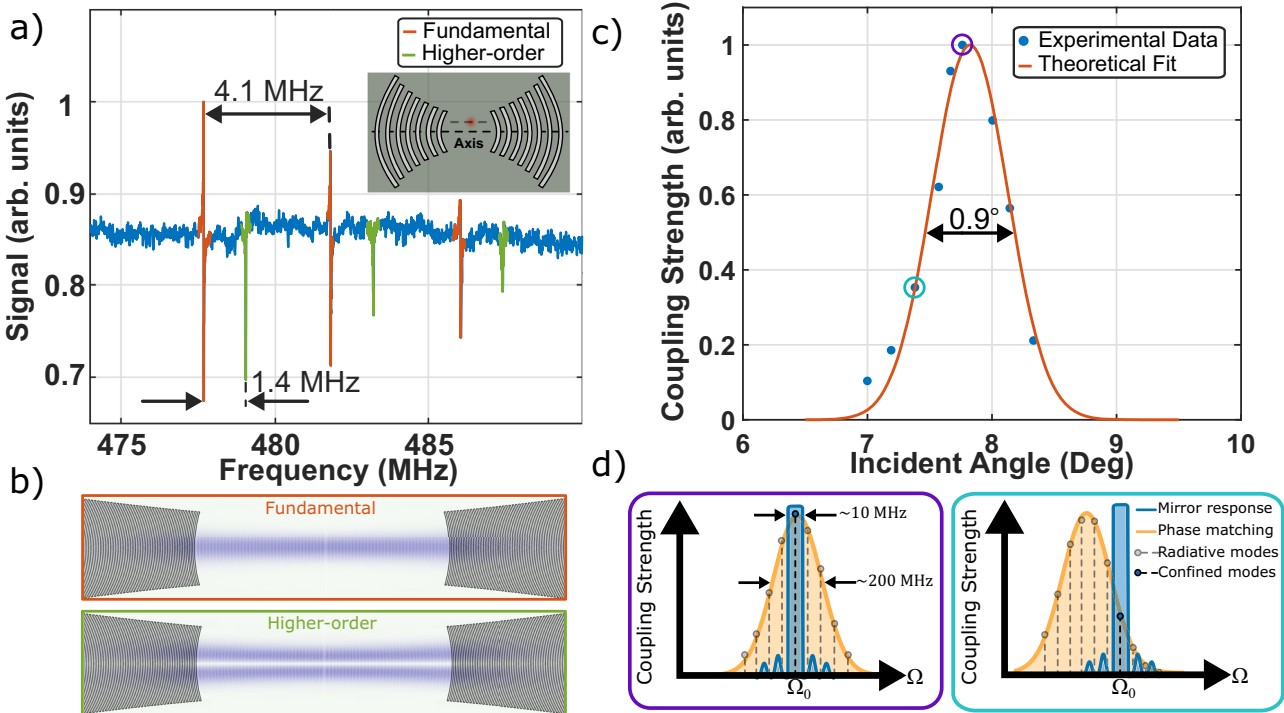

**Fig. 3 | Measured spatial mode spectrum and angular dependence. a** The optomechanical response from higher-order acoustic modes is observed in a [100]-oriented cavity with a mirror spacing of $L \approx 350\,\mu m$ by laterally displacing the optical fields as illustrated in the inset figure. The higher-order frequency spacing of 1.4 MHz, is consistent with the theoretical estimate. **b** Finite element simulations of the corresponding acoustic mode profiles. **c** Optomechanical coupling strength as a function of angle of incidence with a Gaussian fit overlayed. **d** A graphical representation of the positions of the phase-matching envelope relative to the acoustic mirror response (not to scale), for illustration, angles of incidence are indicated with the same color outline as for the respective points in **c**. **d** Left: The phase-matching envelope coincides with the mirror response for maximal opto-mechanical coupling strength. Right: The phase matching envelope is detuned from the mirror-defined SAW mode resulting in a weaker optomechanical response.

## Non-contact probing of SAW cavity dissipation mechanisms

Acoustic dissipation is typically measured using electromechanical techniques, which also include the dissipation from external device structures such as the electrodes, electrical ports, and impendence matching circuits, limiting insights into material and structural dissipation mechanisms. In contrast, the coherent optical interaction is contact-free and not limited by these extrinsic effects. A direct probe into phonon loss mechanisms will be valuable for basic material science as well as for optimizing novel SAW device technologies. Here the coherent optical technique is used to determine the dominant loss mechanisms between SAW propagation and mirror losses for Gaussian resonators and to extract the temperature dependence of the dissipation. The acoustic quality factor is measured as a function of effective cavity length and temperature for cavities in both the [100]-oriented (piezo-inactive) and [110]-oriented (piezo-active) cavities. The measured cavities are all designed to have identical parameters except for the mirror separation, which varies from 150 μm to 500 μm. Since multiple axial modes are observed in each SAW device, the axial mode displaying the highest quality factor is chosen to characterize the loss. The cavity lengths are chosen to minimize effects resulting from optical absorption in the metallic reflectors (See section S9 of Supplementary Information). While the quality factor increases as a function of cavity length for both cavity orientations, the nature of the variation differs for the two cavity orientations as illustrated by the theoretically motivated fits (Fig. 4a, b). For devices oriented along the [100]-direction, the acoustic quality factor at $T = 4\,K$ displays a linear dependence on cavity length (Fig. 4a), suggesting that the losses in SAW cavities primarily occur within the acoustic mirrors through mechanisms such as scattering into the bulk, ohmic losses, and acoustic losses within the reflectors. For devices oriented along the

[110]-direction, the dependence is nonlinear because, while still ~4 times lower than the mirror loss, intrinsic propagation loss is not negligible (see section S8 of Supplementary Information for additional analysis). The dependence of quality factor on temperature is also investigated from $T = 4$–160 K (Fig. 4c, d) for a fixed mirror separation ($L = 500\,\mu m$ for the [100]-oriented cavities and $L = 350\,\mu m$ for the [110]-oriented cavities). The [100]-oriented cavities exhibit a sharp fall and a subsequent plateau at $Q \sim 20,000$ within the measured temperature range. The observed trend suggests that while the [100]-oriented cavities are mirror-loss limited at low temperatures ($T < 70\,K$), at high temperatures ($T > 70\,K$) the quality factor is primarily determined by intrinsic propagation losses which exhibit weak temperature dependence at high-temperatures[66–69]. In contrast, the [110] cavities exhibit a linear decrease of $Q$ with temperature. This decrease could be attributed to increasing ohmic losses, which, in turn, result from the increasing resistivity of metallic reflectors[70]. Previous measurements of Gaussian SAW cavities on GaAs without the potential for ohmic losses in the mirrors (through superconducting reflectors[71] as well as non-metallic reflectors[62]) demonstrating large acoustic quality factors ($2 \times 10^4$) suggest that the losses observed in the [110]-oriented, piezo-active,cavities primarily result from ohmic losses within the metallic reflectors. Additional insights could be derived from temperature-dependent quality factor measurements at additional cavity lengths including longer lengths where the effects of mirror loss are reduced, from cavities where ohmic losses are reduced such as superconducting-mirror cavities, as well as from alternative cuts and material types. Importantly, because of the non-contact nature of coherent optical coupling, these measurements directly reflect intrinsic device properties, as opposed to details of the probe, providing a rich source of information across a wide range of relevant SAW

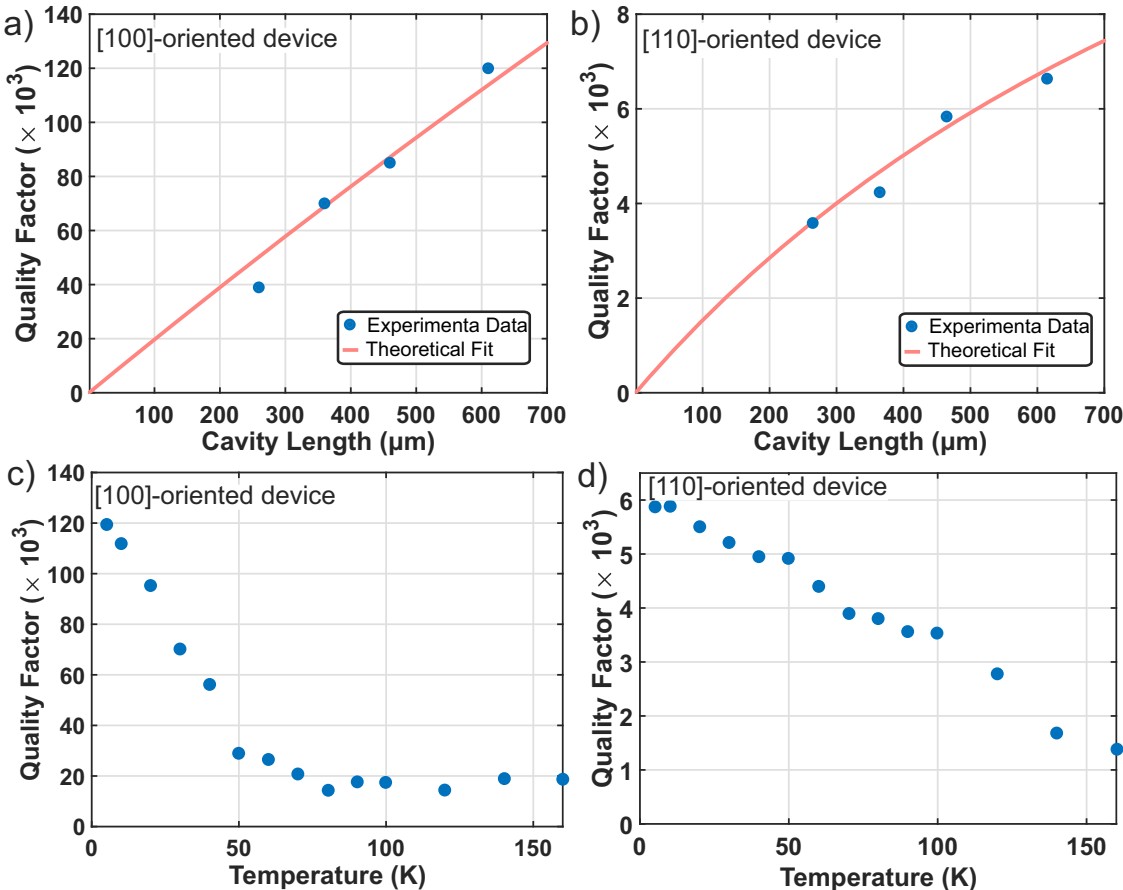

**Fig. 4 | Characterizing loss in SAW devices.** Quality factor as a function of cavity length for cavities oriented along the (**a**) [100] and (**b**) [110] directions at $T = 4$ K. While [100]-oriented devices have a linear dependence of quality factor on cavity length corresponding to cavity loss primarily dominated by the acoustic mirrors, [110]-oriented devices and have a nonlinear dependence on cavity length resulting from larger intrinsic propagation losses along this direction. Quality factor as a function of temperature for cavities oriented along the (**c**) [100] and (**d**) [110] direction. The two orientations display qualitatively distinct dependencies, suggesting differences in acoustic loss mechanisms.

device parameters. This is illustrated in section S9 of Supplementary Information as well where the effects of optical absorption are clearly delineated from those of electrostriction through controlled measurements.

## Discussion

This report introduces a powerful new coherent optomechanical platform in which two non-collinear optical fields parametrically couple through surface acoustic modes of Gaussian SAW cavities. The platform offers high power handling capabilities, requires minimal fabrication, and enables a contact-free piezo-electricity independent coupling to SAW devices enabling record-high quality factor devices in GaAs crystalline substrates. From the results presented here, there are several directions in which specific metrics of interest for applications can be improved. For example, the principles outlined here can be readily applied for the coherent optical coupling of SAW cavity devices with frequencies of several GHz by changing the optical angle of incidence (see section S10 of Supplementary Information). The optomechanical coupling rate of devices can also be improved significantly through reduced acoustic mode volumes in cavities with smaller acoustic waists (see section S10 of Supplementary Information). Moreover, because the acoustic mode volume of the Gaussian SAW cavities scale inversely with the acoustic frequency, GHz SAW cavities naturally offer increased coupling strengths. Acoustic cavity losses can also be further improved by adopting etched groove reflectors in favor of metallic strips to eliminate both ohmic losses within the reflectors

on piezoelectric substrates as well as additional acoustic losses within the reflectors.

A natural extension of the technique presented here would be to enclose the system within optical cavities. A SAW-mediated cavity optomechanical system with an operation frequency of ~4 GHz, Q-factors well exceeding $10^5$, and appreciable cavity-optomechanical coupling rates ($\frac{g_0^c}{2\pi} \sim 1$ kHz) could be achieved through straightforward improvements detailed in Supplementary Information section S10. The power-handling capability of this system (see section S13 of Supplementary Information), limited only by material damage, allows for large intracavity photon numbers ($n_c > 10^9$) which can consequently enable large optomechanical cooperativities ($C_{om} > 100$) (see section S10 of Supplementary Information). This platform therefore yields the high-power handling capability of bulk optomechanical systems[42,72] while also offering large coupling rates, small sizes, and ideal integrability to quantum systems and sensing devices.

A SAW cavity-optomechanical platform may have several straightforward applications. Strain fields of surface acoustic phonon modes can be readily coupled to a range of qubit systems, including spin qubits, quantum dots, and superconducting qubits. Combining SAW cavity-optomechanical platforms with established SAW phonon-qubit coupling schemes, such as Gaussian SAW cavities embedded with qubits[10,14,15], could enable novel all-optical quantum transduction strategies. Optical coupling to several other strain-sensitive quantum systems, including superfluids and 2D materials, can also be realized, which could yield new fundamental insights into novel condensed

matter phenomena. The SAW-based cavity optomechanical system could also serve as an alternate platform for microwave-to-optical transduction schemes circumventing conventional challenges such as poor phonon-injection efficiencies, low-power handling capabilities, and fabrication challenges[7,49,50]. Beyond novel devices for quantum systems, the demonstrated techniques and devices also represent an attractive strategy for realizing a new class of non-contact all-optical SAW-based sensors with targets ranging from small molecules to large biological entities including viruses and bacteria, without electrical contacts or constraints. Moreover, in contrast to prior electro-mechanical techniques, the material versatility available to the opto-mechanical coupling presented enables broadly applicable material spectroscopy for basic studies of phonons and material science, including for non-transparent and opaque materials such as metals and semiconductors.

In summary, here we demonstrate coherent optical coupling to surface acoustic cavities on crystalline substrates. A novel non-collinear Brillouin-like parametric interaction accesses high-frequency Gaussian SAW cavity modes without the need for piezo-electric coupling, enabling record cavity quality factors. Optomecha-nical coupling in SAW cavities could be enabling for hybrid quantum systems, condensed matter physics, SAW-based sensing, and material spectroscopy. For hybrid quantum systems, this interaction, in con-junction with demonstrated techniques of strong coupling of SAWs to quantum systems (e.g. qubits, 2D materials, and superfluids), could form the basis for the next generation of hybrid quantum platforms. For sensing, this platform could enable a new class of SAW sensors agnostic to piezoelectric properties and free of electrical constraints and resulting parasitic effects. Finally, the coherent coupling techni-que enables detailed phonon spectroscopy of intrinsic mechanical loss mechanisms for a wide array of materials without the limitations of extrinsic probing devices.

## Methods

### Numerical methods
Determining the exact acoustic reflector profiles requires the SAW group velocity as a function of the angle from the chosen SAW cavity axis, i.e., the anisotropy of the substrate. This is calculated by numerically solving acoustic wave equations with appropriate boundary conditions. To efficiently confine SAW fields, the shape of the reflector must match the radius of curvature of the confined Gaussian mode. The calculated group velocity can then be used to determine the radius of curvature of the reflectors as a function of the axial location and angle from the cavity axis ($R(x,\theta)$). These reflector profiles are imported into finite element software to validate the cavity designs by verifying the stability of high-Q Gaussian-like SAW modes (Fig. 1d–f). A detailed description of the FEM simulation procedure is provided in section S3 of Supplementary Information.

### Device fabrication
To fabricate the GaAs devices, a single crystal [100]-cut GaAs is coated with a PMMA polymer layer and the required reflector profiles are drawn on the polymer with an e-beam lithography tool. Subsequently, the required thickness of metal, in this case 200 nm Aluminum, is deposited using an ultra-high vacuum e-beam evaporation tool sys-tem. Finally, the excess polymer is removed using an acetone bath to obtain the experimental devices. A more detailed description of the device fabrication is provided in section S4 of Supplementary Information.

### Temperature control
The SAW devices are mounted to a custom copper mount which is secured to the base plate of a liquid-helium closed-cycle cryostat (Montana Instruments s50). The temperature is monitored and con-trolled through a calibrated thermometer secured to the mount.

### Phonon spectroscopy
A sensitive phonon-mediated four-wave mixing measurement techni-que is developed, building off of related techniques for measuring conventional Brillouin interactions. The SAW cavity mode is driven with two optical tones, which are incident at angles designed to target specific phonon frequencies. A probe beam at a disparate wavelength which is incident collinear to one of the drive tones, scatters off the optically driven SAW cavity mode to generate the measured response. The angle of incidence of the optical fields is controlled through off-axis incidence on a well-calibrated aspheric focusing lens (see section S6 of Supplementary Information). The optomechanically scattered signal is collected on a single-mode collimator and spectrally filtered using a fiber-Bragg grating to reject excess drive light. The resulting signal is combined with a local oscillator (LO) and measured with a balanced detector (see section S5 of Supplementary Information). The measured signal is a coherent sum of frequency-independent Kerr four-wave mixing in the bulk of the crystalline substrate and the optomechanical response, giving rise to Fano-like resonances. This spectroscopy technique can resolve optomechanical responses with <fW optical powers. A detailed description of the experimental appa-ratus and the angle tuning technique is provided in sections S5 and S6 of Supplementary Information, respectively. Additional controlled experiments are performed to verify that the optomechanical response (e.g., phonon frequency and quality factor) is independent of ambient light and optical power variations up to 1.2 W, to confirm the absence of any potential nonlinear absorption or carrier generation effects. These power-dependent measurements also confirm that unwanted optical reflections (e.g., from the back surface of the sub-strate) and scattering within the cryostat chamber do not induce excess heating of the SAW substrate.

## Data availability
Data underlying the results presented in this paper are not publicly available at this time but may be obtained from the authors upon request.

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

## Acknowledgements
This work was supported by The National Science Foundation (ECCS-1943658, WHR), The Office of Naval Research (N00014-20-1-2424, JMN and WHR; N00014-23-1-2704, WHR), The Defense Advanced Research Projects Agency (D23AP00169-00, WHR), The Army Research Office (W911NF-19-1-0167, JMN), and the University of Rochester (WHR and JMN). We thank the University of Rochester Integrated Nanosystems Center for their expertize and assistance with fabrication of SAW devices.

## Author contributions
A.I. and W.H.R conceived, designed, and planned the experiment with assistance from J.M.N.; Y.P.K. and J.M.N. designed and fabricated the SAW devices with assistance from A.I.; A.I. performed measurements and analyzed experimental data with assistance from W.X.; A.I. developed the optomechanical coupling model with assistance from W.X.; W.H.R. supervised the research and all authors participated in writing the manuscript.

## Competing interests
The authors declare no competing interests.
