## [Peer Review File · Nature Communications]

Coherent Optical Coupling to Surface Acoustic Wave DevicesREVIEWER COMMENTS

Reviewer #1 (Remarks to the Author):

Recommendation

The paper presents a novel approach for excitation and probing surface acoustic waves (SAWs) modes on a planar resonator, that could be used, in principle, even on non-piezoelectric substrates. While the results are convincing and this technique can be adopted in optomechanical experiments, in my opinion the authors overestimate its applicability to quantum technologies. This reduces the impact of the work. Therefore, unless a revision addresses the loss mechanics possibly introduced by the optical driving itself, I believe the paper does not reach the standards for publication in Nature Communication.

Paper's summary and key results

The paper presents an original work on the excitation and detection of SAWs' modes in a planar resonator realized on GaAs substrate. The authors design, optimize by simulations and fabricate SAW resonators with two orientations (with respect of the crystallographic axis of the substrate) and with mode frequencies around 500MHz. They demonstrated coherent excitation of resonant SAWs' modes by Brillouin optomechanical interaction with two optical beams shined in the middle of the cavity. They measure the mechanical modes' quality factor and estimate their optomechanical coupling from the experimental parameters. Finally, they apply this novel measurement technique to the study of loss in their resonators.

General comments

1. In the conclusion the authors suggest that this measurement technique can be used to couple the strain field with a qubit, enabling novel transduction strategies. However, I have a few concerns regarding the impact on the quantum community due to three aspects:

a) While it is true that the interdigitate (IDT) launcher within the cavity produces some loss, this is the coupling scheme for all superconducting architectures. In addition, the coupling is mediated by the electric field connected to the strain field. Could the authors be more specific on the way they envision the coupling of a qubit to SAW?

b) The experiment is carried out for SAWs frequencies of 500MHz. Except for the work in [2], usually systems in circuit QED use resonant interaction with qubits or other resonators (above 4 GHz). This will ensure a passive reset of the SAW cavity state (by the cold thermal bath). While the authors present a possible extension of their cavities to the GHz, they assume no degradation of performance.

c) The beams (stokes and pump fields) shine directly in the cavity. While the authors comment on the possibility of having a much more robust thermal balance, I am not convinced that this experiment can be carried out in continuous wave (CW) in a dilution cryostat (the usual cooling power of the mixing chamber is tens of uW) without warming the chip.

Could the authors comment on these three points?

2. I also wonder why the quality factors of cavities oriented along [110] and [100] differ so much (almost two orders of magnitude). According to the authors, this is due to ohmic loss in the

reflectors. However, if this is the case, the quality factor of the resonator oriented along the direction [110] should be strongly dominated by mirror loss (and the substrate loss should be negligible). Therefore, going from cavity length of 150 to 500um the quality factor should increase 3 times (formula 100 in the supplementary), while it only increases 2 times. On the other hand, the quality factor is indeed 3 times for the resonators oriented along [100], showing that this is mainly limited by mirror loss. This makes me conclude that the reason for such a difference in the two directions is not in the Ohmic loss.

Is it possible to predict or extrapolate the quality factor by the simulation (even for smaller fingers in the mirrors)? Is this result in agreement with the experiment?

The gallium arsenide has a direct bandgap (1.4eV corresponding to 870nm). Excited electrons will dissipate energy due to SAW travelling in the “piezo active” direction. Was the experiment performed in the absence of visible light?

I understand the main novelty of this work is in the coupling scheme, but I would like to be sure that this technique is not detrimental to the SAW resonator’s performance.

3. In several parts of the paper the authors say that this coupling scheme is efficient. Nevertheless, I do not find any definition of what they consider efficient or any quantitative estimation. I think it would be very important to mention or estimate it (for example photon-phonon conversion efficiency \times at y mW Stoke power and z mW pump).

4. All the study was performed for the first mirror stopband. But with this technique should be possible to access other stopbands. If my understanding is correct, this is another advantage of the technique. In any case, I suggest the author mention if this is possible or not in the manuscript or in the supplementary information.

Detailed review

Abstract

In the sentence “The non-contact nature...” I suggest removing “record-high”, or at least explaining that this value is comparable to literature for very small mode volume resonators. In literature, many SAWs resonators achieved larger quality factors at higher frequencies (see [1] $Q_i \approx 450k$ at 2.4GHz, [2] $Q_i \approx 200k$ at 2GHz).

Main

I suggest to reshape the sentence: “In addition, the absence of interdigital...”. In fact, in both ref. [2] (on GaAs) and [3] the quality factor is larger, for higher frequencies and for electrodes within the cavities. Moreover, I believe it is important to state in the introduction that this value is achieved at cryogenic temperatures (4K).

Non-Collinear Brillouin-like Optical Coupling to Surface Acoustic Waves

In the sentence containing: “...Brillouin scattering from bulk acoustic waves” I would include also the reference in [4] where this type of scattering was observed in SAWs.

In figure 1d, I suggest to indicate or mention for which y the “z-cross section” was plotted.

Optomechanical spectroscopy of SAW cavities

In the sentence: “Optomechanical measurements are made for” I suggest adding “in a cryogenic

environment" if this is the case.

In figure 2a and 2c, are the modes shown in the spectrum all the resonator modes? Why do the different modes have different heights? Why the Signal to Noise Ratio is different in a and c? I suggest the authors to add this information.

In figure 2b and 2d, what function was used for the fit?

Regarding the fQ products, the authors mention that their device is $6e13$ Hz, comparable to the best electromechanical SAW device. To my knowledge the best value reported is in [1] with $1.1e15$ Hz, and [2] with $4e14$. I suggest adjusting the sentence accordingly.

Non-contact probing of SAW Cavity Dissipation Mechanism

In figure 4a and 4b, it is not clear if these quality factors are measured in a cryogenic environment or not. If this is the case this should be clearly mentioned in the text.

In figure 4c and 4d do all the resonators and resonators' modes show the same trend?

The resonators used to compile figure 4 are all multimode. How was the particular mode chosen? How was the temperature on the chip monitored? These are technical aspects, but I believe having such details is important to assess the feasibility of future experiments with this technique.

Discussion and conclusion

In the sentence "A SAW-mediated cavity..." the authors mentioned that with straightforward improvement it is possible to achieve Q exceeding $1e5$ at 4GHz. Nevertheless, loss in SAW cavities scales with frequency at least linearly (see [2]). I suggest using more conservative number for high-frequency Q , or at least explain why they do not expect any degradation.

Supplementary material

In figure S6, the authors surprisingly see a decrease in frequency with an increase in power when the beam overlaps with the mirror. Was this observed for the piezo-active direction as well? I presume this measurement was performed at 4K. Is this effect also consistent at higher temperatures?

In section S9, it is stated that the overlapping can be reduced to $1e-15$. Was any reflection from the surface considered? Could this produce any possible issue?

[1] R. Manenti, Nature Communications volume 8, 975 (2017)

[2] G. Andersson, npj Quantum Information 7, 15 (2021)

[3] R. Manenti, Phys. Rev. B 93, 041411(R) (2016)

[4] J. Beugnot, Nature Communications, 5, 5242 (2014)

Reviewer #2 (Remarks to the Author):

The manuscript "Coherent Optical Coupling to Surface Acoustic Wave Devices" reports on a surface acoustic wave resonator where the acoustic modes are driven and measured optically via optomechanical interaction. Traditional surface acoustic wave resonators are fabricated on a piezoelectric substrate or non-piezoelectric substrate with piezoelectric film to excite acoustic waves by the electric signal via piezoelectricity. The system the authors report basically does not require piezoelectricity and interdigital transducers in the resonators. The authors also characterize two devices with a piezoelectric-active and piezoelectric-inactive orientation of the GaAs substrate and show the higher quality factor is realized in a piezoelectric-inactive device. It is concluded that the ohmic loss due to the current induced in the metal mirror by the piezoelectric effect contributes significantly to the loss factor.

The techniques presented in this report are applicable to non-piezoelectric materials like silicon, diamond, or sapphire, where complicated fabrication processes are needed to realize the optomechanical system. It would be possibly applicable to superconducting or magnetic materials, which are basically incompatible with piezoelectricity, and fabrication for optomechanical systems is not established. This report extends the physics of optomechanics based on Brillouin scattering from bulk systems to surface acoustic wave devices and enriches the application of surface acoustic wave resonators in various fields. I would recommend publication in Nature Communications after the following points are addressed.

I'm curious if this technique will also be used for non-transparent substrates like silicon and other metallic material, which have been not the scope of the surface acoustic wave devices. If possible, what kind of modifications to the measurement system is needed?

The resonator has fundamental and non-degenerate higher-order transverse modes as shown in Fig. 3a due to the curved mirrors. However, the spectra in Fig. 2a and 2c have only fundamental modes and FSR, and the higher-order modes are not observed. The spectra obtained by this method are sensitive to the optical beam spot position. Even if the beam spot is at a node in anti-symmetric mode (Fig. 1e) like the axis of the resonator, a symmetric higher-order mode (Fig. 1f) should be observed in addition to the fundamental mode. Can the authors present information about the position of the beam spot on the device? Why is the symmetric higher-order mode not observed in this spectrum?

The spectra shown in Fig.2a and Fig.2c are different. Fig.2c shows a simple Lorentzian-like peak but the signal in Fig.2a is asymmetric. I recommend the authors mention the reason and the fitting function for each signal in the main text.

In Fig. 4c, the quality factor in the piezo-inactive resonator shows non-linear temperature dependence. Near 60K, the slope shows the change. The authors consider that the ohmic loss is not dominant in this device and the propagation loss is also not dominant in both devices because the quality factor increase linearly in cavity length. I recommend the authors comment on the mechanism of this temperature dependence and the dominant loss channel in the piezo-inactive resonator.

In S10, the authors discuss the cavity optomechanical system using the surface acoustic wave resonator. In the estimation of coupling the authors suppose the fiber cavity with a small cavity

length of $\sim 10\text{-}100\ \mu\text{m}$. The optical incident beam in this report should have a finite angle to the surface (θ) to satisfy the phase-matching condition. Fiber cavities are used with their surface facing each other, so it would be incompatible with the method in this report. What configuration of the combined system does the author suppose?

Reviewer #3 (Remarks to the Author):

The authors describe an opto-mechanical device in a gallium arsenide substrates that couples between two off-chip optical drive beams and a surface acoustic mode of the substrate. The two beams interfere on the device surface and within the substrate, and stimulate the surface acoustic wave through radiation pressure and surface and volume electrostriction. The surface acoustic mode is confined within a cavity of distributed mechanical reflectors, in the form of deposited metallic gratings. When the frequency difference between the optical drive beams and their angles of incidence are chosen correctly, the optical forces are wavenumber-matched with the surface acoustic mode. The process is observed through photoelastic modulation of a third, optical probe beam, and heterodyne detection of its reflection.

In my humble opinion, this is very good work. It is novel, it is timely, it is significant, it is practical, and it is thoroughly executed and reported. The claims made are very well supported by evidence throughout the main text and supplementary information.

I recommend that this work is published, almost as it is. Please consider the following suggestions.

1. The supplementary information 9 compares between parametric and absorptive stimulation mechanisms. Data is presented in normalized form. Could the authors compare the magnitudes of the two effects? For the same drive powers, which effect is stronger?
2. Following on the same argument, the Fano shapes of Fig. 2b and Fig. S7.a are attributed to the interplay of Kerr effect and opto-mechanics. I tend to accept this explanation, but it is given very briefly. This point can be elaborated upon.
3. The Fano shapes were not observed for the piezo-active [110] orientated device (Fig. 2d). Why?
4. In Supplementary equations 36 and 77, could the authors comment which of the three terms is the most significant? Is the stimulation primarily a volume effect or a surface effect?
5. Lastly, the supplementary material document can be edited for better care and clarity (typos, spaces, and such). Please take another look.

We are thankful for the thoughtful Reviewer comments and appreciation of the work's novelty. Through this review process, many of the constructive comments we believe have significantly improved the overall quality of the science and clarity of presentation in the revised manuscript and supplement which we hope is now suitable for publication in Nature Communications.

Reviewer comments are repeated below in bold, with the author comments and corresponding changes to the manuscript and Supplementary Information following in blue font.

Reviewer #1 (Remarks to the Author):

Recommendation

The paper presents a novel approach for excitation and probing surface acoustic waves (SAWs) modes on a planar resonator, that could be used, in principle, even on non-piezoelectric substrates. While the results are convincing and this technique can be adopted in optomechanical experiments, in my opinion the authors overestimate its applicability to quantum technologies. This reduces the impact of the work. Therefore, unless a revision addresses the loss mechanics possibly introduced by the optical driving itself, I believe the paper does not reach the standards for publication in Nature Communication.

We thank the reviewer for their thorough and thoughtful review; we address these comments in detail below.

Paper's summary and key results

The paper presents an original work on the excitation and detection of SAWs' modes in a planar resonator realized on GaAs substrate. The authors design, optimize by simulations and fabricate SAW resonators with two orientations (with respect of the crystallographic axis of the substrate) and with mode frequencies around 500MHz. They demonstrated coherent excitation of resonant SAWs' modes by Brillouin optomechanical interaction with two optical beams shined in the middle of the cavity. They measure the mechanical modes' quality factor and estimate their optomechanical coupling from the experimental parameters. Finally, they apply this novel measurement technique to the study of loss in their resonators.

General comments

1. In the conclusion the authors suggest that this measurement technique can be used to couple the strain field with a qubit, enabling novel transduction strategies. However, I have a few concerns regarding the impact on the quantum community due to three aspects:

a) While it is true that the interdigitate (IDT) launcher within the cavity produces some loss, this is the coupling scheme for all superconducting architectures. In addition, the coupling is mediated by the

electric field connected to the strain field. Could the authors be more specific on the way they envision the coupling of a qubit to SAW?

We thank the reviewer for this question. Surface acoustic waves indeed enable quantum control of superconducting qubits through IDTs, including for entangling distant qubits, accessing the strong coupling regime and for the generation of non-classical phonon states. However, importantly, many recent studies have demonstrated that SAWs can also strongly couple to most solid-state qubit platforms [1,2] such as quantum dots [3–5], nitrogen/ silicon vacancy centers in diamond [6,7], color centers in SiC [8] and flying electron qubits [9–11]. In addition to solid-state qubits, surface acoustic waves have also been shown to strongly couple to other condensed matter quantum systems namely superfluids [12,13] and 2D-materials [14–16]. In particular, recent works on coupling SAWs to optical quantum dots embedded within Gaussian SAW resonators [3,4] that closely resemble our system helps illustrate the wide range of possibilities that exist for realizing novel hybrid quantum systems interfaced with SAWs.

We envision coupling qubits with SAWs by building from this large body of existing work by introducing, for the first time, a tailorable optical coupling technique. The SAWs in these established systems would effectively act as a quantum bus allowing for coherent exchange of information between the qubits and incident optical fields mediated through acoustic fields. For example, Ref. [8] demonstrates coupling of Gaussian SAWs fields to spin qubits in SiC. Combining this qubit system with our technique could enable addressing these spin qubits with optical fields, as opposed to requiring electrical contacts. Interestingly, the phonon frequencies used in this work are similar to our demonstrations (~ 560 MHz). Note that the technique demonstrated in [8] would not be compatible with platforms such as silicon spin qubits, given the absence of piezoelectricity. Our technique, however, will allow for the implementation of hybrid quantum systems on practically any transparent material. Additionally, optical control of SAWs can be employed in conjunction with electromechanical techniques to construct new microwave-to-optical quantum transduction schema.

We have highlighted relevant portions within the introduction Section of the revised manuscript addressing this point, for additional emphasis. In the discussion Section of the revised main text, we also expand with an example system for Optical-SAW-qubit coupling.

b) The experiment is carried out for SAWs frequencies of 500MHz. Except for the work in [2], usually systems in circuit QED use resonant interaction with qubits or other resonators (above 4 GHz). This will ensure a passive reset of the SAW cavity state (by the cold thermal bath). While the authors present a possible extension of their cavities to the GHz, they assume no degradation of performance.

This is certainly a good point that we appreciate the opportunity to clarify. In general, we agree that performance of SAW cavities can degrade as the frequency increases to several GHz and therefore directly scaling the current design without modification may not achieve record performance. However, as the referee mentions later in their report, previous electromechanical demonstrations have demonstrated intrinsic quality factors exceeding 10^5 at ~ 4 GHz [17,18]. These cavities moreover have geometric sizes closely resembling cavities (\sim mm scale) in this work and have the same substrate (GaAs) and mirror material (Al), suggesting that reaching $> 10^5$ quality factors at high frequencies is possible at ~ 100 mK temperatures. Moreover, since cavities proposed in this work do not have electrodes and

compensate for diffraction losses through spherical acoustic cavity mirrors, optimized high frequency Gaussian SAW cavities (e.g. with an optimized mirror geometry based on etched grooves) may even exceed quality factors of cavities that employ electrodes. Finally, optomechanical interactions demonstrated here can be realized on virtually any substrate and crystallographic direction, regardless of piezoelectricity, allowing for the freedom to select intrinsically low-loss materials and crystallographic axes.

We have added a paragraph to Section S10 of the Supplementary Information to discuss this point in detail.

c) The beams (stokes and pump fields) shine directly in the cavity. While the authors comment on the possibility of having a much more robust thermal balance, I am not convinced that this experiment can be carried out in continuous wave (CW) in a dilution cryostat (the usual cooling power of the mixing chamber is tens of uW) without warming the chip.

To address this important point we present additional analysis here and in the Supplementary Information of the revised manuscript. We can make a fairly simple first-order estimate of how much cooling power would be required at different sample temperatures. To estimate this, the system is assumed to be well coupled to the thermal bath because of the large (e.g. 4mm x 4mm) substrate contact area. This is in contrast to nanoscale/microscale optomechanical devices which are often suspended and have holes (or other features) which, in a high vacuum environment, provide a poor thermal link to the bath. Next, assuming that the dominant source of optical heating is from absorption, the absorbed optical power (P_a) can be expressed as

$$P_a = \alpha L P_{in},$$

where α is the absorption coefficient, L is the length of the substrate, and P_{in} is the incident optical power. For a crystalline GaAs substrate the absorption coefficient for incident optical wavelength of $1.55 \mu m$ and temperature $< 4K$ is given in the literature by $\alpha = 0.01 \text{ cm}^{-1}$ [19]. The substrate length used in this work is $L = 300 \mu m$. To ensure that the system stays at the desired temperature, the optical heating should be less than the cooling power of the cryostat (P_c), i.e. $P_c \geq P_a$ or $P_{in} \leq \frac{P_c}{\alpha L}$. Assuming a cooling power of $P_c = 30 \mu W$ at 10 mK , $P_c = 1000 \mu W$ at $T = 100 \text{ mK}$ and $P_c = 300 \text{ mW}$ at $T = 1 \text{ K}$ (from Oxford Instruments dilution cryostat manuals) the maximum allowable optical powers to prevent sample heating are $P_{in} \approx 100 \text{ mW}$ at $T = 10 \text{ mK}$, $P_{in} \approx 3 \text{ W}$ at $T = 100 \text{ mK}$ and $P_{in} \approx 300 \text{ W}$ at $T = 1 \text{ K}$. Note that these estimations are strongly material dependent. Materials such as quartz and diamond have lower intrinsic optical losses which will enable larger operational powers. Larger operational powers can also be achieved by reducing the substrate thickness. Since acoustic modes penetrate only a few wavelengths into the bulk, this would have negligible effect on optomechanical dynamics.

For additional support, a recent demonstration confirms negligible effect of laser heating ($\sim 1 \text{ W}$ continuous-wave laser) on phonon occupation of bulk acoustic resonators in their quantum ground state ($T \sim 100 \text{ mK}$) [20] for comparable resonator sizes ($\sim \text{mm-cm}$).

A new Section S13 has been added to the revised Supplementary Information to support the high-power handling capabilities of the SAW optomechanical platform.

Could the authors comment on these three points?

2. I also wonder why the quality factors of cavities oriented along [110] and [100] differ so much (almost two orders of magnitude). According to the authors, this is due to ohmic loss in the reflectors. However, if this is the case, the quality factor of the resonator oriented along the direction [110] should be strongly dominated by mirror loss (and the substrate loss should be negligible). Therefore, going from cavity length of 150 to 500um the quality factor should increase 3 times (formula 100 in the supplementary), while it only increases 2 times. On the other hand, the quality factor is indeed 3 times for the resonators oriented along [100], showing that this is mainly limited by mirror loss. This makes me conclude that the reason for such a difference in the two directions is not in the Ohmic loss. Is it possible to predict or extrapolate the quality factor by the simulation (even for smaller fingers in the mirrors)? Is this result in agreement with the experiment?

Again the reviewer makes a fantastic point here which requires a more detailed analysis to address which we have done and describe below. We believe this additional detail is particularly important, useful, and improves the quality of the manuscript. It is also important for us to also emphasize that the primary goal and ultimately the value of our results in this paper is to establish a new coupling mechanism between light and sound. One of the potential impacts of this novel all-optical, non-contact technique of acoustic coupling is the study of acoustic dissipation. In this way it is not our intention to fully disambiguate the many potential acoustic dissipation channels but rather to demonstrate that we have new access to this type of basic information. Given the limitations in the literature (particularly for experiments) in understanding the interplay of various acoustic loss processes, extensively characterizing the loss processes would require considerable research beyond the scope of this manuscript. However, in response to this comment, for the data presented, we have now taken several additional steps in analysis to extract much more information from what was originally presented which we hope further emphasizes the value of the technique.

As the reviewer mentions, the slope of Q vs L for the piezo cavities differs from that observed for the non-piezo cavities. This difference can be explained through the full nonlinear analysis of the quality factor which we present here. We find that a general expression for Quality factor as a function of length (Eq. 98 of Supplementary Information) accounting for both propagation and mirror losses reveals that the presence of modest propagation losses can explain the observed Q vs L dependences and the differences in the two cavity types. The generalized expression of Quality factor Q as a function of cavity length (Eq. 98 of Supplementary Information) is given as

$$Q = \frac{f_0 L}{v_R} \left(\frac{1}{\alpha_p L + \alpha_M} \right),$$

where, f_0 , v_R and L refer to the resonant frequency, the acoustic velocity and the effective cavity length, respectively. α_p and α_M are the corresponding coefficients of propagation and mirror loss, respectively. In contrast to taking the linear limit for negligible propagation loss (Eq. 99 of Supplementary Information), we now numerically fit the experimental results to the full theoretical model resulting in the fitting values listed in Table. R1. Q vs L is now plotted (Fig. 1a-b) with the revised fitting function where L is now more accurately defined as the effective cavity length including the $\sim 120 \mu m$ penetration length into the mirrors.

Parameter	Value (units)	Description
f_0	475 MHz	Acoustic frequency
v_R	2615 m/s	SAW velocity
α_p	$10^{-5} - 10^{-4} \mu\text{m}^{-1}$	Propagation loss
α_M	0.9	Mirror loss

Parameter	Value (units)	Description
f_0	510 MHz	Acoustic frequency
v_R	2880 m/s	SAW velocity
α_p	$8 \times 10^{-3} \mu\text{m}^{-1}$	Propagation loss
α_M	11.2	Mirror loss

Table R1: Estimated loss parameters (α_p, α_M) for piezo-inactive (left) and piezo-active (right) cavities.

Numerical fits agree well with experimental observations for both piezo-inactive (Fig. R1a) and -active (Fig. R1b) cavities. While the fit to the piezo-inactive sample (Fig. R1a) is linear (for the plotted cavity lengths), it is nonlinear for the piezo-active samples, which corresponds to larger acoustic propagation losses (compare α_p in Table. S1). Note that because the piezo-inactive devices are approaching the linear limit where propagation loss is negligible, there is a larger uncertainty in the fit parameter for propagation loss in this case. Additional data points with larger cavity lengths could help refine this

Fig. R1: Revised Q vs L plots for a) piezo-inactive and b) piezo-active cavities. The revised plots have been correctly plotted as a function of total cavity length as opposed to mirror separation. Additionally, observations are now fit to the full Q vs L model accounting for propagation losses as opposed to linear fits. The recovered loss estimates are listed in Table. R1.

estimate. In both cavities the mirror loss dominates propagation loss (by $\sim 100x$ for the piezo-inactive cavities and by $\sim 4x$ for the piezo-active cavities for $L \sim 370 \mu\text{m}$). Both the propagation and mirror losses for the piezo-cavity are approximately $>100x$ ($\alpha_{p,p}/\alpha_{p,np}$) and $10x$ ($\alpha_{M,p}/\alpha_{M,np}$) larger, respectively, than those in piezo-inactive cavities. The larger propagation losses in the piezo-active cavities correspond to the observed reduced slope in our Q vs L plots. While the larger mirror losses are consistent with the presence of ohmic losses along in the piezo-active cavities, the source for the larger propagation losses in the piezo-active cavities is less obvious. The larger loss in the [110] direction is consistent with the fact that SAWs propagating along the [110]-directions are leaky; they are also known as pseudo-surface acoustic waves (as opposed to non-leaky SAWs in the [100]-direction). SAWs along this direction intrinsically have a small wavevector component pointing into the bulk of the substrate. This loss is zero exactly along the [110]-direction and is non-zero for directions away from it [21,22]. Since our cavities have focused Gaussian beams with a spread of acoustic wavevectors, the propagation losses could be higher for such cavities which would contribute to the observed dependence of the quality factor on cavity length. As a consequence of acoustic power radiating into the bulk, Intrinsic losses along leaky directions are typically larger than those along non-leaky directions. For example, lithium niobate supports both leaky surface acoustic waves and true SAWs on different material cuts.

The 64° Y-cut of LiNbO_3 supports leaky SAWs along the X-axis, similar to [110]-oriented devices investigated in this work and have an estimated propagation loss of $\sim 0.036 \text{ dB}/\lambda$ [23]. Whereas the 128° Y-cut of the lithium niobate that supports true (non-leaky) SAWs propagating along the X-axis, analogous to [100]-devices investigated in this work, has an approximate propagation loss of $\sim 1.5 \times 10^{-4} \text{ dB}/\lambda$ which is about 2 orders smaller than the cut supporting leaky SAWs.

We note that in our analysis so far, we have assumed that mirror losses (α_M) are independent of cavity lengths. This assumption may begin to lose validity for piezo cavities. Ohmic losses within piezo-active cavities arise from the non-uniform Gaussian acoustic strain profile and resulting non-uniform piezoelectric potential along the metallic reflectors. The ohmic loss produced in any one reflector may vary as a function of local acoustic beam waist ($w_a(x)$) and the local acoustic amplitude ($A(x)$). The shape of acoustic metallic reflectors are designed to match the local wavefronts of the Gaussian acoustic mode which varies as a function of the cavity length and the beam waist at the center of the cavity. Consequently, the cumulative ohmic losses within the mirrors could be different for SAW cavities with different effective lengths. This variation of ohmic losses as a function of exact mirror shapes, local acoustic profile, beam width, etc could additionally contribute to the observed difference within the two classes of SAW cavities investigated in this work.

We also agree with the reviewer's suggestion of using simulation tools to gain insights into these loss processes. However, given that 3D FEM simulations of SAW cavities are time intensive, and computationally expensive, simulation of loss processes would require developing alternate simulation tools. Alternatively, one could isolate these loss processes through additional experiments, for e.g., by employing superconducting metallic reflectors to eliminate ohmic losses or employing etched non-metallic mirrors. We are pursuing fabrication of devices of this kind for future studies, for example.

The Q vs L plots (Fig. 4a, 4b) in the main text are replaced with the more detailed plots presented here. Furthermore, we provide a brief explanation for the observed differences in the dependence of quality factor on length for the two cavities.

Section S8 has been revised to include the analysis presented here.

The gallium arsenide has a direct bandgap (1.4eV corresponding to 870nm). Excited electrons will dissipate energy due to SAW travelling in the "piezo active" direction. Was the experiment performed in the absence of visible light? I understand the main novelty of this work is in the coupling scheme, but I would like to be sure that this technique is not detrimental to the SAW resonator's performance.

We would like to thank the referee for pointing out the possible interaction between the charge carriers in the semiconductor with the surface acoustic field and its possible implications on the measured quality factor. In principle it is possible that excess carriers could be generated within the semiconductor bulk and introduce additional acoustic dissipation channels in SAW cavities oriented along the piezo-active directions. In response to this comment, we have conducted additional experiments to rule out the effects of ambient light and power dependent absorption on acoustic dissipation.

Ambient light: While the optical fields ($\lambda_o = 1.55 \mu\text{m}$, $E_o = 0.75 \text{ eV}$) used in our experiments are well below the bandgap of the substrate, visible ambient light in and around our experimental setup if incident on the substrate could in principle generate additional carriers. To study this possibility, we measure the quality factor of a piezo-oriented SAW device with the same parameters as in the text with and without ambient light (Fig. R2). The measured resonance frequency ($f_0 = 504.55$ in both cases), quality factor ($Q = 6000$ (ON), $Q = 6100$ (OFF)) and coupling strength in the two cases are well within experimental errors, suggesting that ambient light and the excess carriers produced subsequently have negligible effect on acoustic cavity losses.

Figure R2: Optomechanical response of a SAW device oriented along the [110] piezo-active direction when the ambient light is a) ON and b) OFF.

Intensity dependent absorption: Semiconductors like Si and GaAs display two-photon effects under high optical field intensities. In this case, two photons from the incident fields can be absorbed to generate excess carriers. In integrated silicon photonics systems, for example, two-photon absorption (TPA) limits the amount of power that can be propagated through the waveguides. In our system larger powers could in principle yield stronger TPA resulting in larger induced carrier density that could in turn induce stronger carrier-photon coupling. We therefore rule out intensity dependent effects by measuring the optomechanical spectra of a [110]-oriented device nominally identical to devices characterized in the paper as function of input acoustic drive powers from 300 mW to 1200 mW. This experiment is identical to the experiments performed to distinguish parametric and absorptive effects described in the Supplement. We observe no discernable changes in acoustic resonance properties for acoustic drive powers in the range 300 mW to 1.2 W (Fig. R3a-R3e). The quality factor and resonance frequency (Fig. R3f) show little change as a function of input drive powers, suggesting that the incident

optical fields do not induce appreciable TPA and consequently do not introduce additional acoustic losses (even for the reasonably large optical powers investigated in this work).

Figure R3: Optomechanical response of a SAW device oriented along the [110] piezo-active direction as function of input acoustic power for input optical drive power of a) 300 mW, b) 500 mW, c) 700 mW, d) 1000 mW and e) 1200 mW. f) Quality factor and resonance frequency as a function of acoustic drive power.

The conclusions detailed here are summarized in the revised Methods section of the main text to clarify that the interaction is not dependent on power changes or ambient light.

3. In several parts of the paper the authors say that this coupling scheme is efficient. Nevertheless, I do not find any definition of what they consider efficient or any quantitative estimation. I think it would be very important to mention or estimate it (for example photon-phonon conversion efficiency x at y mW Stoke power and z mW pump).

We agree that our definition of ‘efficient’ was not clearly defined in the paper. Our primary intention here is to emphasize that the optomechanical coupling in our system is by far the strongest demonstrated with SAWs on a planar cavity. To quantify phonon-photon coupling strengths, we have used two normalized metrics: the optomechanical coupling rate (g_0) and the Brillouin gain coefficient (G_B). Of the two metrics, the Brillouin gain naturally lends itself to a definition suggested by the reviewer. The fractional stokes power ($\Delta P_S/P_S$) scattered by the optomechanical process can be expressed as

$$\frac{\Delta P_S}{P_S} = G_B P_p.$$

For cavities demonstrated in this work, the Brillouin gain is approximately $G_B \sim 3 \times 10^{-5} W^{-1}$ and we anticipate these to be at least an order of magnitude better for high frequency low-mode volume cavities as described in the supplement Section 10. For comparison, previous works on optically accessed SAWs on planar cavities [24] have been restricted to low frequencies ($\Omega < 100$ MHz), low

quality factors ($Q \sim 100$) and low travelling-wave coupling rates ($g_0^T \sim 2\pi \times 50 \text{ Hz}$). Therefore, compared to typical parameters for other systems, e.g. as given in [24] ($g_0^T \sim 2\pi \times 50 \text{ Hz}$, $Q \sim 100$, and $\Omega \sim 100 \text{ MHz}$), our efficiency (g_0 (travelling – wave) $\sim 2\pi \times 1.5 \text{ kHz}$, $Q \sim 120,000$, $\Omega \sim 500 \text{ MHz}$), is approximately 10^5 times larger than prior works. This increase is a result of stronger phase matched interactions, smaller acoustic mode volumes, and significantly larger quality factors.

In Section S1 of the revised Supplementary Information, we have expanded on the definitions provided for G_B and g_0 to provide additional context regarding the efficiency of the interaction.

4. All the study was performed for the first mirror stopband. But with this technique should be possible to access other stopbands. If my understanding is correct, this is another advantage of the technique. In any case, I suggest the author mention if this is possible or not in the manuscript or in the Supplementary Information.

We agree and this is a great advantage of this system that we did not discuss. There is no restriction to optically coupling (phase matching) to cavity modes in other stopbands. However, unsurprisingly, these modes generally have lower Q-factors.

We have now modified Section S3 of the Supplementary Information to briefly discuss this additional novelty of this new technique.

Detailed review

Abstract

In the sentence “The non-contact nature...” I suggest removing “record-high”, or at least explaining that this value is comparable to literature for very small mode volume resonators. In literature, many SAWs resonators achieved larger quality factors at higher frequencies (see [1] $Q_i \approx 450\text{k}$ at 2.4GHz , [2] $Q_i \approx 200\text{k}$ at 2GHz).

We agree with the reviewer and have revised the abstract to specify that this work demonstrates access to record-high quality factors for small mode volume cavities.

The abstract of the revised main text has now been modified to specify that record-high quality factors are observed for small mode volumes.

Main

I suggest to reshape the sentence: “In addition, the absence of interdigital...”. In fact, in both ref. [2] (on GaAs) and [3] the quality factor is larger, for higher frequencies and for electrodes within the cavities. Moreover, I believe it is important to state in the introduction that this value is achieved at cryogenic temperatures (4K).

While we agree with the reviewer that other works have demonstrated SAW cavities with higher quality factors, electrodes have been well established to introduce additional losses in the form of undesirable scattering and ohmic losses. Our technique therefore is unique in that it can achieve quality factors that can be understood to be material/ mirror geometry limited, with no consequence from the IDT; this doesn't however mean that the quality factors we measure are always the highest for any cut or

material...etc, as the reviewer points out. That being said, the quality factor measured in this work, to the best of our knowledge, is the largest quality factor measured on any substrate within a focused (Gaussian) SAW cavity. Focused cavities moreover are desirable for achieving the smallest mode-volumes and ultimately strong coupling to optical and other quantum systems.

The suggested sentence in the introduction of the main text has been modified in the revised manuscript to emphasize that the presented Q-factors are achieved for cryogenic temperatures and small-mode volume (Gaussian) cavities.

Non-Collinear Brillouin-like Optical Coupling to Surface Acoustic Waves

In the sentence containing: "...Brillouin scattering from bulk acoustic waves" I would include also the reference in [4] where this type of scattering was observed in SAWs.

We agree the literature pointed out by the referee is important and relevant.

We have now put this reference into context in the Introduction of the revised manuscript.

In figure 1d, I suggest to indicate or mention for which y the "z-cross Section" was plotted.

Fig. 1d is plotted for $y=0$, i.e. along the acoustic cavity axis. We have described this in the revised caption.

We have added in the revised caption that the XZ cross-Section plotted is for $y = 0$.

Optomechanical spectroscopy of SAW cavities

In the sentence: "Optomechanical measurements are made for" I suggest adding "in a cryogenic environment" if this is the case.

We thank the reviewer for their suggestion.

This sentence has been modified as suggested to include the cryogenic conditions.

In figure 2a and 2c, are the modes shown in the spectrum all the resonator modes? Why do the different modes have different heights? Why the Signal to Noise Ratio is different in a and c? I suggest the authors to add this information.

Fig. 2a and 2c show all SAW modes which yield an optomechanical signal above the noise floor of the experimental apparatus. The longitudinal modes display disparate relative signal strengths because they have different acoustic quality factors. The measured optomechanical signal scales as $\sim Q^2$ and as a result the different longitudinal modes have different relative heights. The different acoustic quality factors are consistent with their position relative to the center frequency of the acoustic mirror stopband (Fig. 1c). Modes with acoustic frequency at to the center of the stop band frequency could be confined better and hence have larger Q-factors. Quality factors (frequency) of the longitudinal SAW resonances shown in Fig. 2a are- 3.5×10^4 (477.3 MHz), 1.2×10^5 (479.5 MHz), 8.3×10^4 (481.8 MHz), 7.6×10^4 (484.2 MHz), 7.2×10^4 (486.5 MHz), 2.5×10^4 (488.7 MHz). Notice that while the four strongest resonances display similar quality factors (within 70 % of the maximum),

the two resonances on either ends of the measured frequency range, as expected, show drastically lower quality factors. Dependence of quality factor on the spectral position of the resonance relative to the mirror stopband has been observed in previous electromechanical studies as well [3,4].

Since the optomechanical strength scales with the quality factor, the signal strength observed in piezo-active samples with $Q \sim 6500 - 7000$ is approximately 50-100x weaker than signals observed in the non-piezo samples resulting in lower SNR.

The main text has been revised to describe the cause of unequal signal strengths for different longitudinal modes. Low signal-to-noise ratio for signals observed in piezo-active active cavities has also been addressed in the revised manuscript.

In figure 2b and 2d, what function was used for the fit?

For all the resonances observed, a Fano-like fitting function was used to extract the relevant resonance parameters. The exact functional form for the measured signal power as a function of frequency from a Fano-like resonance can be expressed as [25]-

$$P_s = P_{nl} \left| e^{i\phi} + \frac{D_m \Omega_m / (2Q_m)}{\Omega_m - \Omega - i\Omega_m / (2Q_m)} \right|^2 + P_{bg},$$

where Ω_m and Q_m refer to the resonant frequency and quality factor of an acoustic resonance with index m , P_{nl} is the power from frequency independent nonlinear processes, D_m is the relative strength between the Brillouin and frequency independent interactions, and ϕ_m is the relative phase between the resonant Brillouin interaction and the coherent non-resonant background signal (e.g., from the Kerr nonlinearity), and P_{bg} accounts for residual incoherent background noise. The experimentally measured scattered power as a function of the drive frequency, Ω , is given by $P_s^{ex}(\Omega)$. This resonance data can be fit to give parameters $p_m = \{D_m, \phi_m, \Omega_m, Q_m, P_{nl}, P_{bg}\}$ such that-

$$P_s^{ex}(\Omega) = P_s(\Omega, p_m).$$

To provide readers with this information, we have described this fitting technique in the revised text and referenced the relevant optomechanical literature. We have also added a description of the deviation in spectral response shapes between the two measurements. Additionally, we have appended an additional Section (Section S12) in the Supplementary Information describing our fitting procedure.

Regarding the fQ products, the authors mention that their device is 6e13 Hz, comparable to the best electromechanical SAW device. To my knowledge the best value reported is in [1] with 1.1e15 Hz, and [2] with 4e14. I suggest adjusting the sentence accordingly.

We agree with the reviewer that these works have demonstrated higher fQ products and we have modified the sentence to suggest that this work has fQ products comparable to best small mode volume SAW cavities.

Non-contact probing of SAW Cavity Dissipation Mechanism

In figure 4a and 4b, it is not clear if these quality factors are measured in a cryogenic environment or not. If this is the case this should be clearly mentioned in the text.

We thank the reviewer for pointing this out. All the quality factors in Fig. 4a and 4b were measured at a temperature of 4K.

This Section and caption now explicitly state the temperature at which the experiments were conducted.

In figure 4c and 4d do all the resonators and resonators' modes show the same trend? The resonators used to compile figure 4 are all multimode. How was the particular mode chosen?

This is a good question that given the time-consuming nature of temperature measurements is beyond the scope of this study. There are a total of ~30 axial modes in our system. However, we can say that indirect observation of neighboring modes in all of our Brillouin studies suggest that the other longitudinal modes will display similar trends. Said differently, we have not yet seen any mode deviate from these nominal trends from the selection of measurements we have made in this space. We choose the axial SAW mode with the largest quality factor to characterize acoustic loss as a function of length and temperature.

In the revised manuscript we now explicitly state that the mode with the highest Q-factor is chosen to study temperature and length dependent effects.

How was the temperature on the chip monitored? These are technical aspects, but I believe having such details is important to assess the feasibility of future experiments with this technique.

While the temperature of the sample itself was not directly monitored, the temperature of the metallic mount on which the sample was mounted to thermally couple to the cryostat was monitored.

An additional Section has been added to the revised Methods Section to specify how temperature is controlled and monitored.

Discussion and conclusion

In the sentence "A SAW-mediated cavity..." the authors mentioned that with straightforward improvement it is possible to achieve Q exceeding $1e5$ at 4GHz. Nevertheless, loss in SAW cavities scales with frequency at least linearly (see [2]). I suggest using more conservative number for high-frequency Q, or at least explain why they do not expect any degradation.

We agree with the reviewer that quality factor of SAW devices can degrade at higher frequency and that current devices demonstrated in the work, without optimization, would not have the high quality factors quoted in the outlook Section. However, as the reviewer pointed out in their summary of this work, previous studies have demonstrated SAW cavities with acoustic quality factors exceeding 4×10^5 at $\Omega \sim 3$ GHz on GaAs. Moreover, as described in the manuscript, we foresee significant optimization of our cavity geometry (for e.g. mirror geometry). Combined with the fact that our cavities do not have

parasitic loss mechanisms resulting from electrodes and diffraction suggest that once our devices are optimized to minimize loss, quality factors of 10^5 should be accessible.

Section S10 of Supplementary Information has now been revised to further justify quality factors $\sim 10^5$ when estimating the performance of high frequency cavities.

Supplementary material

In figure S6, the authors surprisingly see a decrease in frequency with an increase in power when the beam overlaps with the mirror. Was this observed for the piezo-active direction as well? I presume this measurement was performed at 4K. Is this effect also consistent at higher temperatures?

While perhaps surprising, these frequency shifts are at least straight-forward to explain. Acoustic velocities of most crystalline materials typically increase at lower temperature. This is because elastic constants for most crystals depend inversely on temperature [26]. Consequently, the cavity frequency also increases at lower temperatures. We believe the following mechanism therefore describes our measurements: optical absorption in the acoustic mirrors causes local thermal heating which increases the effective temperature of the SAW cavity resulting in a decrease in cavity frequency. This is supported by our temperature dependent measurements, where we observe a decrease in cavity frequency as a function of increasing temperature. Cavities oriented along the piezo-active direction also display the same dependence of cavity resonance frequency on temperature and we expect the piezo-active cavity frequency to decrease with increased overlap with the acoustic mirrors.

Section S9 of the revised Supplementary Information has been revised to provide an explanation for the changes in resonant frequency as a function of temperature.

In Section S9, it is stated that the overlapping can be reduced to $1e-15$. Was any reflection from the surface considered? Could this produce any possible issue?

The original calculation did not consider reflections from the back surface that could potentially overlap with the absorbing mirrors. The original calculation treats the straight-forward contribution to the absorption from spatial overlap of the incident Gaussian beam with beam diameters of $\sim 60 \mu m$ and the metallic gratings (Fig. R4a). For acoustic mirror separation lengths larger than $\sim 350 \mu m$ the optical power (assuming ideal Gaussian beams) overlapping with the mirror is negligible $< 10^{-15}$ (i.e. from $1 - \operatorname{erf}\left(\frac{175 \mu m}{30 \mu m}\right)$). If there were reflections from the back surface or from other scattering sources that in addition, overlapped with the

Figure R4: a) A $30 \mu m$ Gaussian beam incident on the SAW device with mirrors separated by $\sim 500 \mu m$. b) Acoustic drives incident on the SAW device intersect at the surface, after which they deviate away from each other as they specularly reflect from the back surface of the substrate.

mirrors, two unwanted effects could result, absorptive driving of phonons and thermal heating of the substrate. We briefly explore these possibilities below.

The incident optical fields in the center of the SAW cavity will specularly reflect from the back surface of the substrate. However, since the two optical fields subtend opposite angles with respect to the surface normal, the spatial overlap between them will reduce as the optical beams eventually move away from each other (Fig. R4b). Given the small beam sizes ($\sim 30 \mu m$) the large separation between the mirrors ($\sim 500 \mu m$), this overlap will be negligibly small. Since an overlap is required for a response, this reflection should not yield a measured response from our probe.

Additionally, the average power incident on the metallic gratings after reflection from the back surface could cause thermal effects, like changes in resonant frequency or quality factor (Section S9 of Supplementary Information). However, one would expect power that, if this is the case, as the incident power is increased, the undesirable absorption would increase as well, resulting in power dependent effects as observed in Section 9 of the Supplementary Information. However, power dependent measurements on both piezo and non-piezo devices, do not reveal power-dependent effects when the optical beams are incident in the center of the SAW device, suggesting that reflections or undesirable scattering do not result in thermal effects.

The Methods Section has been revised to mention that unwanted reflections and scattering do not produce measurable effects.

[1] R. Manenti, *Nature Communications* volume 8, 975 (2017)

[2] G. Andersson, *npj Quantum Information* 7, 15 (2021)

[3] R. Manenti, *Phys. Rev. B* 93, 041411(R) (2016)

[4] J. Beugnot, *Nature Communications*, 5, 5242 (2014)

We thank the reviewer again for their efforts with this review. We believe that the revisions suggested or prompted by the referee have added significant value to the revised manuscript.

Reviewer #2:

The manuscript "Coherent Optical Coupling to Surface Acoustic Wave Devices" reports on a surface acoustic wave resonator where the acoustic modes are driven and measured optically via optomechanical interaction. Traditional surface acoustic wave resonators are fabricated on a piezoelectric substrate or non-piezoelectric substrate with piezoelectric film to excite acoustic waves by the electric signal via piezoelectricity. The system the authors report basically does not require piezoelectricity and interdigital transducers in the resonators. The authors also characterize two devices with a piezoelectric-active and piezoelectric-inactive orientation of the GaAs substrate and show the higher quality factor is realized in a piezoelectric-inactive device. It is concluded that the ohmic loss due to the current induced in the metal mirror by the piezoelectric effect contributes significantly to the loss factor.

The techniques presented in this report are applicable to non-piezoelectric materials like silicon, diamond, or sapphire, where complicated fabrication processes are needed to realize the

optomechanical system. It would be possibly applicable to superconducting or magnetic materials, which are basically incompatible with piezoelectricity, and fabrication for optomechanical systems is not established. This report extends the physics of optomechanics based on Brillouin scattering from bulk systems to surface acoustic wave devices and enriches the application of surface acoustic wave resonators in various fields. I would recommend publication in Nature Communications after the following points are addressed.

We thank the reviewer for their appreciation and thoughtful comments on our work.

I'm curious if this technique will also be used for non-transparent substrates like silicon and other metallic material, which have been not the scope of the surface acoustic wave devices. If possible, what kind of modifications to the measurement system is needed?

We would like to thank the referee for pointing out an important possibility that this technique can be applied to non-transparent materials such as semiconductors (when excited with optical fields with energy larger than their bandgap) and metallic substrates. While the proposed experimental setup can be used, a new theoretical framework will have to be developed to fully understand optomechanical processes mediated by optical absorption. We describe some of these effects in Section S9 of the Supplementary Information. Unlike parametric photoelastic processes, optical absorption will result in additional local thermal heating of the substrate, and these effects will have to be understood to leverage these processes. Additionally, in metallic substrates acoustic effects are accompanied by free electron effects. Indeed, as envisioned by the reviewer, when properly developed, all-optical techniques like ours could offer deep insights into acoustic processes and even perhaps lead to qualitatively new types of coupled photon-phonon-electron dynamics in these materials. We mention these possibilities in the discussion section of the revised manuscript. However, by the virtue of being driven by optical absorption, these processes would inherently be dissipative and could pose challenges when employed for quantum systems, which are well suited for the parametric coherent interactions demonstrated here.

The revised manuscript the discussion Section now briefly discusses the possibility of optically studying phonon physics in opaque (non-transparent) materials.

The resonator has fundamental and non-degenerate higher-order transverse modes as shown in Fig. 3a due to the curved mirrors. However, the spectra in Fig. 2a and 2c have only fundamental modes and FSR, and the higher-order modes are not observed. The spectra obtained by this method are sensitive to the optical beam spot position. Even if the beam spot is at a node in anti-symmetric mode (Fig. 1e) like the axis of the resonator, a symmetric higher-order mode (Fig. 1f) should be observed in addition to the fundamental mode. Can the authors present information about the position of the beam spot on the device? Why is the symmetric higher-order mode not observed in this spectrum?

We thank the reviewer for this question. Transverse acoustic modes of a Gaussian resonator, as in the case of optical cavities, form a mutually orthogonal basis. Therefore, an optical driving that is perfectly optomechanically mode-matched to the fundamental Gaussian mode will yield no overlap with any other acoustic higher-order mode, regardless of the symmetry of the higher-order mode.

However, as suggested by the referee, experimental optical fields are unlikely to *exactly* mode-match to the fundamental Gaussian SAW cavity mode. To address this case, we have performed additional numerical calculations to calculate the optomechanical coupling coefficient of the higher order symmetric Hermite-Gaussian SAW mode when driven by experimental optical fields ($\sim 30 \mu\text{m}$ beam diameter). The summary of the calculations are as follows:

- 1) The higher order acoustic mode by the virtue of having both positive and negative displacements yields smaller acousto-optic overlap with a symmetric optical forcing produced by the optical Gaussian beams. **As a consequence, the Brillouin gain coefficient is reduced by approximately 3x when compared to the optomechanical overlap with the fundamental mode.**

Figure R5 : Absorptive optomechanical interactions in [100]-oriented cavity. a) Optomechanical spectrum over a large range displays two fundamental SAW cavity modes (marked as F) along with the first anti-symmetric higher order mode (HO1) and symmetric higher order mode (HO2). b) A zoom-in of the HO2 resonance.

- 2) The higher order mode also has larger mode volume by **approximately 8x which results in a reduction in Brillouin gain by $\sim 8x$.**
- 3) Consequently, the total Brillouin gain is **reduced by approximately $\sim 24x$** and the measured signal which scales with the square of the **Brillouin gain reduces by $\sim 500x$** . This drastic decrease in the signal strength prevents our experiments from resolving the scattering from these acoustic cavity modes. However, these higher order modes can be accessed through absorptive driving of the SAW modes (which is significantly stronger). One such measurement of a $\sim 150 \mu\text{m}$ length SAW cavity oriented along the [100]-direction is shown below (Fig. R5a) along with a zoom-in of the higher order symmetric mode (Fig. R5b).

The revised main text now briefly discusses the coupling strengths of interactions with higher-order acoustic modes as detailed by the discussion above.

The spectra shown in Fig.2a and Fig.2c are different. Fig.2c shows a simple Lorentzian-like peak but the signal in Fig.2a is asymmetric. I recommend the authors mention the reason and the fitting function for each signal in the main text.

We agree and thank the reviewer for this question. The final measured signal is a coherent interference of frequency independent Kerr-like effects and resonant optomechanical interactions. In the devices investigated in this work, we believe the frequency independent background results from a combination

of $\chi^{(3)}$ -effects and free-carrier effects in the bulk of the substrate. Because these parameters (relative phase and amplitude) can be different for the [100] and [110]-oriented devices, the spectral shape of the [100] and [110] resonances can vary as well. Fano-like resonances have been commonly observed in on chip and fiber-based optomechanical systems [25,27,28]. So much so that, even within the same optomechanical device, acoustic resonances at different frequencies have been observed to have disparate shapes.

The exact functional form for the measured signal power as a function of frequency from a Fano-like resonance can be expressed as

$$P_s = P_{nl} \left| e^{i\phi} + \frac{D_m \Omega_m / (2Q_m)}{\Omega_m - \Omega - i\Omega_m / (2Q_m)} \right|^2 + P_{bg},$$

where Ω_m and Q_m refer to the resonant frequency and quality factor of an acoustic resonance with index m , P_{nl} is the power from frequency independent nonlinear processes, D_m is the relative strength between the Brillouin and frequency independent interactions, and ϕ_m is the relative phase between the resonant Brillouin interaction and the coherent non-resonant background signal (e.g., from the Kerr nonlinearity), and P_{bg} accounts for residual incoherent background noise. The experimentally measured scattered power as a function of the drive frequency, Ω , is given by $P_s^{ex}(\Omega)$. This resonance data can be fit to give parameters $p_m = \{D_m, \phi_m, \Omega_m, Q_m, P_{nl}, P_{bg}\}$ such that

$$P_s^{ex}(\Omega) = P_s(\Omega, p_m).$$

To provide readers with this information, we have described this fitting technique in the revised text and referenced the relevant optomechanical literature. We have also added a description of the deviation in spectral response shapes between the two measurements. Additionally, we have appended an additional Section (Section S12) in the Supplementary Information describing our fitting procedure.

In Fig. 4c, the quality factor in the piezo-inactive resonator shows non-linear temperature dependence. Near 60K, the slope shows the change. The authors consider that the ohmic loss is not dominant in this device and the propagation loss is also not dominant in both devices because the quality factor increases linearly in cavity length. I recommend the authors comment on the mechanism of this temperature dependence and the dominant loss channel in the piezo-inactive resonator.

We thank the reviewer for emphasizing this point which can be addressed through previous studies. While it is challenging to quantitatively account for all possible loss mechanisms and their relative interplay as a function of temperature in a given device, the essential qualitative trend observed matches well with known dependencies of Rayleigh SAWs. SAW propagation losses which primarily occur from acoustic viscosity effects go as $\sim T^4$ at low temperatures ($< 25 K$) before an intermediate transition region ($25 K - 70 K$), and a temperature independent region at high temperatures ($> 70 K$) [29–32]. Previous experimental and theoretical results (Fig. R6a-b) on SAW propagation losses as a function of temperature on crystalline quartz show good qualitative agreement with our measurements (Fig. R6c). We observe a shallower dependence at low temperature, however, because

at low temperatures, non-piezo SAW devices are mirror-loss dominated and the observed trend is an interplay between mirror losses and propagation losses. At high temperatures our results suggest that cavities are dominated by propagation losses which are largely temperature independent, consistent with previous results. In addition to these loss processes, other loss processes including acoustic scattering from surface roughness, thermal expansion of metallic electrode spacing and subsequent changes to the mirror stopband can also be expected to influence quality factors as a function of temperature.

In stark contrast to the piezo-inactive samples, measurements of the piezo-active samples (Fig. 4d of main text) suggest that they remain mirror loss limited throughout the investigated temperature range because the ohmic losses within these cavities increase with the increasing resistivity of Al metallic electrodes with temperature [33]. Propagation losses can also be expected to increase simultaneously. In both systems, quantitatively disentangling these effects with additional experiments may be a broad and fruitful research direction enabled by this approach.

Theory (Budreau, Carr, 1971) Exp (Budreau, Carr, 1971)

Figure R6 : a) Theoretical and b) experimental results reported in Budreau, Carr, 1971. Previous works observe a steep ($\sim T^4$) dependence at low temperatures, and temperature independent loss at high frequency ($> 70 K$). c) Results observed in this work is in good qualitative agreement with previous works and display a similar transition at $\sim 80 K$, beyond which the quality factor does not strongly depend on temperature.

Section 4 of the revised manuscript now includes an explanation of the change in the dissipation mechanism and the resulting change in the slope of Fig. 4c as discussed in depth above. Several references to previous works are also provided for additional context.

In S10, the authors discuss the cavity optomechanical system using the surface acoustic wave resonator. In the estimation of coupling the authors suppose the fiber cavity with a small cavity length of ~ 10 - $100 \mu\text{m}$. The optical incident beam in this report should have a finite angle to the surface (θ) to satisfy the phase-matching condition. Fiber cavities are used with their surface facing each other, so it would be incompatible with the method in this report. What configuration of the combined system does the author suppose?

We thank the reviewer for raising this important question. It is true that fiber cavities of this type demonstrated so far have surfaces facing each other and adapting fiber-based cavities to SAW-based optomechanical devices will require additional experimentation. We are pursuing this line of research ourselves, currently. Two example SAW-based cavity optomechanical configurations which could be compatible with fiber cavities are:

- 1) A bow-tie style fiber cavity in which four fiber ports act as a single optical resonator. The crossing of the beams at the center, which is determined by the relative distances and curvatures of the mirrors can be adapted to match the phase matched angle (Fig. R7a).

- 2) Two separate cavities, one which confines the optical pump and another which confines the sideband of interest. This configuration allows for simultaneously having both pump and sideband resonant with the optical cavity (Fig. R7b).

Conventional fiber cavities are typically constructed with standard single mode optical fibers which have a cladding diameter of $125\ \mu\text{m}$, which could prevent other fiber ends from being placed close enough to realize the required small optical mode volumes. This possible issue can be circumvented by employing reduced cladding fibers which have a cladding diameter of $40\ \mu\text{m}$. In this case by simple geometric estimate even with the largest 45-degree cavity alignment the fiber limits the cavity length in Fig. R7b to $<100\ \mu\text{m}$, which implies that this geometry does not limit the cavity length (Fig. R7c).

While cavities of these types have not been investigated so far, we believe a SAW-based optomechanical with its unique set of advantages could motivate the development of alternate geometries for low mode volume fiber cavities.

Figure R7 : Possible optical cavity geometries compatible with SAW processes could include a) bow-tie style and b) two separate cavities. c) First-order geometric estimation of cavity length (L_c) for a given fiber diameter (d) at angle of incidence of 45° can be expressed as $L_c \approx 2d$.

Section S10 of the revised supplement has been updated to address the fact that current optical fiber cavities are incompatible with SAW processes without suitable redesign as described above.

Reviewer #3 (Remarks to the Author):

The authors describe an opto-mechanical device in a gallium arsenide substrates that couples between two off-chip optical drive beams and a surface acoustic mode of the substrate. The two beams interfere on the device surface and within the substrate, and stimulate the surface acoustic wave through radiation pressure and surface and volume electro-striction. The surface acoustic mode is confined within a cavity of distributed mechanical reflectors, in the form of deposited metallic gratings. When the frequency difference between the optical drive beams and their angles of incidence are chosen correctly, the optical forces are wavenumber-matched with the surface acoustic mode. The process is observed through photoelastic modulation of a third, optical probe beam, and heterodyne detection of its reflection.

In my humble opinion, this is very good work. It is novel, it is timely, it is significant, it is practical, and it is thoroughly executed and reported. The claims made are very well supported by evidence throughout the main text and Supplementary Information.

I recommend that this work is published, almost as it is. Please consider the following suggestions.

We appreciate the referee's thoughtful comments and appreciation that the work "is novel, it is timely, it is significant, it is practical, and it is thoroughly executed and reported".

Below, we address specific comments raised by the referee.

1. The Supplementary Information 9 compares between parametric and absorptive stimulation mechanisms. Data is presented in normalized form. Could the authors compare the magnitudes of the two effects? For the same drive powers, which effect is stronger?

We thank the reviewer for raising this important question. Owing to the limitations of the experimental scheme employed in the work, absorption processes are difficult to characterize quantitatively. This is because we do not have independent control over the position of the acoustic drive fields and probe fields, i.e. the spatial position of the probe beam is locked to the position of the drive fields. Consequently, the probe undergoes undesirable scattering from the metallic reflectors which in turn prevents us from accurately characterizing the coupling strength (G_a) of absorption mediated optomechanical processes. Additionally, the strength of the absorptive scattering is observed to be a function of the position of the optical beams with respect to the acoustic mirror, which is challenging to accurately measure. However, by measuring the sideband power for absorptive processes for fixed given optical drive powers, we can estimate a lower bound (other positions on the mirror could possibly yield stronger absorptive effects) on the coupling strength of the absorptively mediated optomechanical process. The equivalent Brillouin coupling coefficient of the absorptive process (G_a) is found to be larger than that of the parametric process (G_p) by at least 10x, i.e. $\frac{G_a}{G_p} > 10$.

In the revised Supplementary Information, Section S9 has been updated to discuss the relative strength of the absorptive and parametric processes.

2. Following on the same argument, the Fano shapes of Fig. 2b and Fig. S7.a are attributed to the interplay of Kerr effect and opto-mechanics. I tend to accept this explanation, but it is given very briefly. This point can be elaborated upon.

We agree and thank the referee for the comment. This interplay is important and noticeable, and fortunately it has been observed in several prior publications studying Brillouin-based bulk interactions.

We have now elaborated on the observation of Fano-like resonances with references to previous publications showing similar effects in the revised manuscript and an additional Section S12 has been added to the supplementary to describe this interplay between Kerr-like and optomechanical effects.

3. The Fano shapes were not observed for the piezo-active [110] orientated device (Fig. 2d). Why?

The exact shape of the Fano-like resonance depends on the phase difference between the background arising from non-resonant processes like Kerr-like nonlinearity and the relative strengths of the two processes. We believe these parameters are different for the SAW devices oriented along the [100] and [110] directions. Fano-like resonances similar to Fig. 2b have been routinely observed in travelling-wave optomechanical devices. Even within the same device acoustic resonances at different acoustic frequencies display distinct Fano-like asymmetric resonances corresponding to different relative phases

between the resonant optomechanical process and frequency independent Kerr-like processes, suggesting that the relative phase is frequency dependent. In addition to conventional Kerr effects our semiconductor samples also support free-carrier mediated Kerr-like effects, which could possibly have different response along piezo and non-piezo directions further contributing to the observed difference in the resonant response. Additional work into understanding the nature of the dominant Kerr-like process would be required to quantitatively understand and explain the observed differences. This is an important area for future investigation. In the revised text, we acknowledge and elaborate on the observed differences in the resonance spectra of the two samples and provide several references to previous works.

To provide readers with this information, we have described this fitting technique in the revised text and referenced the relevant optomechanical literature. We have also added a description of the deviation in spectral response shapes between the two measurements. Additionally, we have appended an additional Section (Section S12) in the Supplementary Information describing our fitting procedure.

4. In Supplementary equations 36 and 78, could the authors comment which of the three terms is the most significant? Is the stimulation primarily a volume effect or a surface effect?

The three forcing mechanisms namely, radiation pressure, surface electrostriction, and bulk electrostriction and their contribution to the gain is a function of material properties such as the relevant photo-elastic coefficient (p_{12}, p_{13} , etc), refractive index (n) and other acoustic parameters characterizing the surface acoustic wave (η, ϕ, γ , etc). For GaAs, the contribution to the acoustic-optic overlap from the three forces is approximately equal. This is a result of the large refractive index and relatively strong photoelastic constants of the chosen substrates. In other materials such as Silicon, which have weaker photoelastic effects, radiation pressure force's contribution to the total acousto-optic overlap would dominate. Inversely, in materials like quartz which have smaller optical indices, photoelastic processes would dominate the parametric interaction.

Section S1 of the revised supplemental information now includes this discussion of the relative strengths of the three optical forcing terms in addition to newly added relative normalized acousto-optic overlap contribution variables ($\alpha_{rp}, \alpha_{es}, \alpha_{eb}$).

5. Lastly, the supplementary material document can be edited for better care and clarity (typos, spaces, and such). Please take another look.

We thank the reviewer for bringing this to our attention. The revised Supplementary material has been significantly edited for typos and issues of clarity.

We would like to thank all of the referees for their comments and suggestions which we believe have significantly improved the revised manuscript and Supplementary Information.

1. P. Delsing, A. N. Cleland, M. J. A. Schuetz, J. Knörzer, G. Giedke, J. I. Cirac, K. Srinivasan, M. Wu, K. C. Balram, C. Bäuerle, T. Meunier, C. J. B. Ford, P. V. Santos, E. Cerda-Méndez, H. Wang, H. J. Krenner, E. D. S. Nysten, M. Weiß, G. R. Nash, L. Thevenard, C. Gourdon, P. Rovillain, M. Marangolo, J. Y. Duquesne, G. Fischerauer, W. Ruile, A. Reiner, B. Paschke, D. Denysenko, D. Volkmer, A. Wixforth, H. Bruus, M. Wiklund, J. Reboud, J. M. Cooper, Y. Q. Fu, M. S. Brugger, F. Rehfeldt, and C. Westerhausen, "The 2019 surface acoustic waves roadmap," *J. Phys. D: Appl. Phys.* **52**, 353001 (2019).
2. M. J. A. Schuetz, E. M. Kessler, G. Giedke, L. M. K. Vandersypen, M. D. Lukin, and J. I. Cirac, "Universal quantum transducers based on surface acoustic waves," *Phys. Rev. X* **5**, (2015).
3. R. A. Decrescent, Z. Wang, P. Imany, R. C. Boutelle, C. A. McDonald, T. Autry, J. D. Teufel, S. W. Nam, R. P. Mirin, and K. L. Silverman, "Large Single-Phonon Optomechanical Coupling between Quantum Dots and Tightly Confined Surface Acoustic Waves in the Quantum Regime," *Phys. Rev. Appl.* **18**, 034067 (2022).
4. P. Imany, P. Imany, P. Imany, Z. Wang, Z. Wang, R. A. DeCrescent, R. C. Boutelle, C. A. McDonald, C. A. McDonald, T. Autry, S. Berweiger, P. Kabos, S. W. Nam, R. P. Mirin, K. L. Silverman, and K. L. Silverman, "Quantum phase modulation with acoustic cavities and quantum dots," *Opt. Vol. 9, Issue 5*, pp. 501-504 **9**, 501–504 (2022).
5. M. Weiß, J. B. Kinzel, F. J. R. Schülein, M. Heigl, D. Rudolph, S. Morkötter, M. Döblinger, M. Bichler, G. Abstreiter, J. J. Finley, G. Koblmüller, A. Wixforth, and H. J. Krenner, "Dynamic acoustic control of individual optically active quantum dot-like emission centers in heterostructure nanowires," *Nano Lett.* **14**, 2256–2264 (2014).
6. S. Maity, L. Shao, S. Bogdanović, S. Meesala, Y. I. Sohn, N. Sinclair, B. Pingault, M. Chalupnik, C. Chia, L. Zheng, K. Lai, and M. Lončar, "Coherent acoustic control of a single silicon vacancy spin in diamond," *Nat. Commun.* 2020 111 **11**, 1–6 (2020).
7. D. A. Golter, T. Oo, M. Amezcua, I. Lekavicius, K. A. Stewart, and H. Wang, "Coupling a Surface Acoustic Wave to an Electron Spin in Diamond via a Dark State," (n.d.).
8. S. J. Whiteley, G. Wolfowicz, C. P. Anderson, A. Bourassa, H. Ma, M. Ye, G. Koolstra, K. J. Satzinger, M. V. Holt, F. J. Heremans, A. N. Cleland, D. I. Schuster, G. Galli, and D. D. Awschalom, "Spin–phonon interactions in silicon carbide addressed by Gaussian acoustics," *Nat. Phys.* **15**, 490–495 (2019).
9. R. Ito, S. Takada, A. Ludwig, A. D. Wieck, S. Tarucha, and M. Yamamoto, "Coherent Beam Splitting of Flying Electrons Driven by a Surface Acoustic Wave," *Phys. Rev. Lett.* **126**, 070501 (2021).
10. S. Hermelin, S. Takada, M. Yamamoto, S. Tarucha, A. D. Wieck, L. Saminadayar, C. Bäuerle, and T. Meunier, "Electrons surfing on a sound wave as a platform for quantum optics with flying electrons," *Nat.* 2011 4777365 **477**, 435–438 (2011).
11. S. Takada, H. Edlbauer, H. V. Lepage, J. Wang, P. A. Mortemousque, G. Georgiou, C. H. W. Barnes, C. J. B. Ford, M. Yuan, P. V. Santos, X. Waintal, A. Ludwig, A. D. Wieck, M. Urdampilleta, T. Meunier, and C. Bäuerle, "Sound-driven single-electron transfer in a circuit of coupled quantum rails," *Nat. Commun.* 2019 101 **10**, 1–9 (2019).
12. H. Byeon, K. Nasyedkin, J. R. Lane, L. Zhang, N. R. Beysengulov, R. Loloee, and J. Pollanen, "Anomalous Attenuation of Piezoacoustic Surface Waves by Liquid Helium Thin Films," *J. Low*

- Temp. Phys. **195**, 336–342 (2019).
13. H. Byeon, K. Nasyedkin, J. R. Lane, N. R. Beysengulov, L. Zhang, R. Loloee, and J. Pollanen, "Piezoacoustics for precision control of electrons floating on helium," Nat. Commun. 2021 **121**, 1–7 (2021).
 14. R. Peng, A. Ripin, Y. Ye, J. Zhu, C. Wu, S. Lee, H. Li, T. Taniguchi, K. Watanabe, T. Cao, X. Xu, and M. Li, "Long-range transport of 2D excitons with acoustic waves," Nat. Commun. **13**, 1–7 (2022).
 15. R. Fandan, J. Pedrós, and F. Calle, "Exciton-Plasmon Coupling in 2D Semiconductors Accessed by Surface Acoustic Waves," ACS Photonics **8**, 1698–1704 (2021).
 16. R. Fandan, J. Pedrós, F. Guinea, A. Boscá, and F. Calle, "Effect of quasiparticle excitations and exchange-correlation in Coulomb drag in graphene," Commun. Phys. 2019 **21**, 2, 1–9 (2019).
 17. G. Andersson, S. W. Jolin, M. Scigliuzzo, R. Borgani, M. O. Tholén, J. C. Rivera Hernández, V. Shumeiko, D. B. Haviland, and P. Delsing, "Squeezing and Multimode Entanglement of Surface Acoustic Wave Phonons," PRX Quantum **3**, 010312 (2022).
 18. G. Andersson, A. L. O. Bilobran, M. Scigliuzzo, M. M. de Lima, J. H. Cole, and P. Delsing, "Acoustic spectral hole-burning in a two-level system ensemble," npj Quantum Inf. 2021 **71**, 7, 1–5 (2021).
 19. M. D. Sturge, "Optical Absorption of Gallium Arsenide between 0.6 and 2.75 eV," Phys. Rev. **127**, 768 (1962).
 20. H. M. Doleman, T. Schatteburg, R. Benevides, S. Vollenweider, D. Macri, and Y. Chu, "Brillouin optomechanics in the quantum ground state," (2023).
 21. M. R. Melloch and R. S. Wagers, "Propagation loss of the acoustic pseudosurface wave on (Zxt)45°GaAs," Appl. Phys. Lett. **43**, 1008–1009 (1983).
 22. W. D. Hunt and B. J. Hunsinger, "A precise angular spectrum of plane-waves diffraction theory for leaky wave materials," J. Appl. Phys. **64**, 1027–1032 (1988).
 23. K. Yamanouchi and M. Takeuchi, "Applications for piezoelectric leaky surface waves," IEEE Symp. Ultrason. **1**, 11–18 (1990).
 24. A. Okada, F. Oguro, A. Noguchi, Y. Tabuchi, R. Yamazaki, K. Usami, and Y. Nakamura, "Cavity Enhancement of Anti-Stokes Scattering via Optomechanical Coupling with Surface Acoustic Waves," Phys. Rev. Appl. **10**, 1 (2018).
 25. H. Shin, W. Qiu, R. Jarecki, J. A. Cox, R. H. Olsson, A. Starbuck, Z. Wang, and P. T. Rakich, "Tailorable stimulated Brillouin scattering in nanoscale silicon waveguides," Nat. Commun. **4**, (2013).
 26. R. Tarumi, K. Nakamura, H. Ogi, and M. Hirao, "Complete set of elastic and piezoelectric coefficients of α -quartz at low temperatures," J. Appl. Phys. **102**, (2007).
 27. A. Iyer, W. Xu, J. E. Antonio-Lopez, R. A. Correa, and W. H. Renninger, "Ultra-low Brillouin scattering in anti-resonant hollow-core fibers," APL Photonics **5**, (2020).
 28. W. H. Renninger, H. Shin, R. O. Behunin, P. Kharel, E. A. Kittlaus, and P. T. Rakich, "Forward Brillouin scattering in hollow-core photonic bandgap fibers," New J. Phys. **18**, 025008 (2016).
 29. H. J. Maris, "Attenuation of Ultrasonic Surface Waves by Phonon Viscosity and Heat Conduction,"

- Phys. Rev. **188**, 1308 (1969).
30. P. J. King and F. W. Sheard, "Viscosity tensor approach to the damping of Rayleigh waves," J. Appl. Phys. **40**, 5189–5190 (1969).
 31. A. A. Maradvdn and A. D. L. Mills, *Calculation of the Anharmonic Damping of Rayleigh Surface Modes** (1968).
 32. A. J. Budreau and P. H. Carr, "Temperature dependence of the attenuation of microwave frequency elastic surface waves in quartz," Appl. Phys. Lett. **18**, 239–241 (1971).
 33. P. D. Desai, H. M. James, and C. Y. Ho, "Electrical Resistivity of Aluminum and Manganese," J. Phys. Chem. Ref. Data **13**, 1131–1172 (1984).

REVIEWER COMMENTS

Reviewer #1 (Remarks to the Author):

Recommendation

The authors addressed almost all my comments and they replied to my main concerns. While I am still not convinced about few of the arguments (as I explain in the following), I think that after mirror revision this work can be disseminated in Nature Communications.

Review revised manuscript

Regarding the newly added section S13, I believe that the estimated 100mW optical power at 10mK is an overestimation of at least 2 orders of magnitude. I think that the dominant source of thermal heating of the sample is indirect heating of the mixing chamber, therefore not due to absorption of the chip, but most likely scattering of the photons not recollected by the measurement setup.

Supposing that most of the light sent to the sample (let's say, very conservatively, 99% of the pump and the Stokes field) is collected and guided away from the mixing chamber if the total injected power is >1mW the cooling power of the cryostat will not be sufficient. I would suggest considering this argument and revising the power handling estimations that the authors reported in section S13.

For completeness, to my knowledge, the highest cooling power for Oxford instrument dilution cryostat is probably <10uW at 10mK, and >25uW at 20mK. This is not a small detail because at 500MHz the thermal population is 10% at 10mK and 43% at 20mK.

Finally, in reference [20] of the authors' reply, 1W is circulating power in the cavity, so the maximum driving power is still in the range of 100s of uW.

Minor Comments

- The x label in fig. 2 d is misspelled
- The x label in fig. S7 b is misspelled

Reviewer #2 (Remarks to the Author):

The authors have addressed all my comments.

The actual optical beam spot size on the device $r_0 \sim 30 \mu\text{m}$ cannot be found in the main text, which seems to be written in only supplementary information. This scale can be compared with the acoustic wavelength, acoustic Gaussian mode distribution, and resonator size. This will help readers understand the experimental situation and data, especially for coupling to higher-order mode. I recommend adding it to the main text.

Reviewer #3 (Remarks to the Author):

The authors addressed the comments I had. This is a good, solid work, and I recommend that it is published.

Reviewer comments are repeated below in bold, with author responses and corresponding changes to the manuscript and Supplementary Information following in blue font.

REVIEWER COMMENTS

Reviewer #1 (Remarks to the Author):

Recommendation

The authors addressed almost all my comments and they replied to my main concerns. While I am still not convinced about few of the arguments (as I explain in the following), I think that after mirror revision this work can be disseminated in Nature Communications.

We appreciate the referee's additional consideration and support of the manuscript and we address this remaining concern in detail below.

Review revised manuscript

Regarding the newly added section S13, I believe that the estimated 100mW optical power at 10mK is an overestimation of at least 2 orders of magnitude. I think that the dominant source of thermal heating of the sample is indirect heating of the mixing chamber, therefore not due to absorption of the chip, but most likely scattering of the photons not recaptured by the measurement setup. Supposing that most of the light sent to the sample (let's say, very conservatively, 99% of the pump and the Stokes field) is collected and guided away from the mixing chamber if the total injected power is >1mW the cooling power of the cryostat will not be sufficient. I would suggest considering this argument and revising the power handling estimations that the authors reported in section S13.

For completeness, to my knowledge, the highest cooling power for Oxford instrument dilution cryostat is probably <10uW at 10mK, and >25uW at 20mK. This is not a small detail because at 500MHz the thermal population is 10% at 10mK and 43% at 20mK.

Finally, in reference [20] of the authors' reply, 1W is circulating power in the cavity, so the maximum driving power is still in the range of 100s of uW.

We agree completely that scattering is an additional and important contribution to heating of the chamber and thanks to this comment, we have now included an analysis of this effect to complement the analysis of absorptive heating. Our intention through this simple analysis is to highlight the fundamental advantages of the SAW optomechanical system when compared to other micro and nanomechanical optomechanical platforms, which is that because we are using a uniquely simple geometry, the contributions from both scattering and absorption can be lower than traditional systems. For optical scattering, the fraction of optical power scattered from a material surface characterized by an RMS (root-mean squared) surface roughness, σ , can be determined through a standard scattering analysis as $S = S_F + S_B = R_0 \left(\frac{4\pi\sigma \cos \theta_i}{\lambda} \right)^2 + T_0 \left(\frac{2\pi\sigma}{\lambda} (n - 1) \right)^2$ [1-4]. Here the total scattering coefficient (S), is expressed as a sum of back scattering (S_B) and forward scattering (S_F) coefficients. R_0 , T_0 , θ_i , λ and n are the scatter-free reflection coefficient, scatter free transmission coefficient, the angle of incidence, the

optical wavelength, and the refractive index respectively. The commercially polished GaAs substrates used in this work have surface roughness of $\sigma = 0.4 \text{ nm}$ [5] (this could be as low as 0.1 nm [6]). The fraction of scattered optical power for an incident wavelength of $\lambda = 1.5 \mu\text{m}$ and incident angle of $\theta_i = 8^\circ$ is then $S \approx 4 \times 10^{-5}$ ($\sim 0.004 \%$) (and as low as $S = 1 \times 10^{-6}$ for $\sigma = 0.1 \text{ nm}$). The low scattering estimate is primarily a result of the large optical wavelength of operation (1550 nm) and low surface RMS roughness. These scattering estimates are consistent with experimental works measuring scattering from smooth optical components, such as lenses, which are typically of the order of $\sim 10^{-5}$ (i.e. tens of ppm) [7,8]. For a more conservative estimate, we assume a scattering coefficient an order of magnitude larger than the numerical estimates ($S \sim 4 \times 10^{-4}$), to account for additional scattering within the bulk of the material or other exceptionally strong, but spurious scattering events. We also conservatively assume that all the scattered power is subsequently absorbed within the chamber and must be cooled. This simple analysis suggests that fundamentally a very low amount of scattered power is required with this system as a result of using only pristine single crystal smooth substrates, which is one of the primary merits of this platform.

Including both scattering and absorption, and using the reviewers numbers for cooling power (specific references included in the revised SI), to ensure that the system stays at the desired temperature, the maximum allowable optical powers to prevent sample heating are $P_{in} \approx 13 \text{ mW}$ at $T = 10 \text{ mK}$, $P_{in} \approx 1 \text{ W}$ at $T = 100 \text{ mK}$ and $P_{in} \approx 100 \text{ W}$ at $T = 1 \text{ K}$. Indeed, by now including scattering this is much lower than the previous estimate and we are thankful for the reviewer in pointing this out. Nonetheless, this power should be higher than it would be with traditional nanofabricated or non-ideal material systems.

Note that for scattering, while other device structures such as acoustic mirrors and lenses, as well as the crystal's back surface could induce additional scattering, these can be mitigated through straight-forward improvements including anti-reflective coatings and larger mirror separations to decrease spatial overlap with acoustic mirrors. Our intention is to convey that still by controlling for this, in stark contrast, other micro- and nano-optomechanical systems have lithographically defined nanostructures (holes, suspended structures, etc.), typically released using chemical etching processes and have poorer surface quality at the level of $\sigma \approx 5 \text{ nm}$ [9], resulting in increased scattering losses, in addition to thermal effects resulting from optical absorption.

We also agree that in reference number 20 in our previous response [10] $\sim 1 \text{ W}$ is the circulating power within the optical cavity, and driving power would be likely be $\sim 100 \mu\text{W}$. All powers used in this section also refer to intra-cavity powers- the total power inside the system. We have now clarified this in the revision.

Section S13 of the Supplementary Information has been rewritten to include the scattering analysis presented here, in addition to optical absorption analysis presented previously.

Minor Comments

-The x label in fig. 2 d is misspelled

This has now been corrected in the revised main text.

-The x label in fig. S7 b is misspelled

This has been corrected in the revised supplementary information.

Reviewer #2 (Remarks to the Author):

The authors have addressed all my comments.

The actual optical beam spot size on the device $r_0 \sim 30 \mu\text{m}$ cannot be found in the main text, which seems to be written in only supplementary information. This scale can be compared with the acoustic wavelength, acoustic Gaussian mode distribution, and resonator size. This will help readers understand the experimental situation and data, especially for coupling to higher-order mode. I recommend adding it to the main text.

The revised main-text now explicitly specifies the optical beam spot on the SAW device.

Reviewer #3 (Remarks to the Author):

The authors addressed the comments I had. This is a good, solid work, and I recommend that it is published.

We appreciate the referee's appreciation of the manuscript.

Finally, we would like to reiterate our gratitude to all the reviewers whose suggestions and recommendations have significantly improved the revised manuscript and Supplementary Information. We hope it is now suitable for publication in Nature Communications.

1. T. V. Vorburger, E. Marx, and T. R. Lettieri, "Regimes of Surface Roughness Measurable With Light Scattering," *Appl. Opt.* **32**, 3401–3408 (1993).
2. H. E. Bennett and J. O. Porteus, "Relation Between Surface Roughness and Specular Reflectance at Normal Incidence," *J. Opt. Soc. Am.* **51**, 123 (1961).
3. P. Beckmann and A. Spizzichino, *The Scattering of Electromagnetic Waves from Rough Surfaces* (1987).
4. S. Schröder, S. Gliech, and A. Duparré, "Measurement system to determine the total and angle-resolved light scattering of optical components in the deep-ultraviolet and vacuum-ultraviolet spectral regions," *Appl. Opt.* **44**, 6093–6107 (2005).
5. M. Corporation, "No Title," <https://www.mtixtl.com/GaAs-Un-100305S2-VGF-1-2-1-1-1.aspx>.
6. Edmund optics, "SUPERPOLISHED OPTICS," <https://www.edmundoptics.com/knowledge-center/trending-in-optics/superpolished-optics/>.
7. E. Collett, *Optical Scattering* (2009).
8. E. M. Capote, A. Gleckl, J. Guerrero, M. Rezac, R. Wright, and J. R. Smith, "Measurements of Optical Scatter Versus Annealing Temperature for Amorphous Ta₂O₅ and TiO₂:Ta₂O₅ Thin Films," **5**, 7–9 (2020).

9. G. Arregui, R. C. Ng, M. Albrechtsen, S. Stobbe, C. M. Sotomayor-Torres, and P. D. García, "Cavity Optomechanics with Anderson-Localized Optical Modes," *Phys. Rev. Lett.* **130**, 1–7 (2023).
10. H. M. Doeleman, T. Schatteburg, R. Benevides, S. Vollenweider, D. Macri, and Y. Chu, "Brillouin optomechanics in the quantum ground state," (2023).

REVIEWERS' COMMENTS

Reviewer #1 (Remarks to the Author):

The authors addressed my remaining comments regarding potential scattering effects; Appendix s13 is now technically correct and I believe that the paper is now suitable for publication in Nature Communications. However, I remain concerned that the authors' estimation still overlooks the primary source of heating of the mixing chamber stage. Specifically, they only consider the scattering at the sample's surface, without assuming any experimental imperfections of the "cryogenic optical setup".

For instance, they assume 100% recollection efficiency for the lasers, yet even a slight deviation from this value will completely saturate the cryostat cooling power. To clarify my statement, the light needs to be guided out of the mixing chamber: if it passes through a small view window, it will not be completely transmitted. The light scattered back will be confined in the cryogenic environment and eventually absorbed.

I still believe this is an important point, since the focus on cryogenic quantum applications, however, I cannot exclude that future improvements in optical setups will overcome such limitations.

The Reviewer comments are repeated below in bold, with the author comments and corresponding changes to the manuscript and Supplementary Information following in blue font.

REVIEWER COMMENTS

Reviewer #1 (Remarks to the Author):

The authors addressed my remaining comments regarding potential scattering effects; Appendix s13 is now technically correct and I believe that the paper is now suitable for publication in Nature Communications. However, I remain concerned that the authors' estimation still overlooks the primary source of heating of the mixing chamber stage. Specifically, they only consider the scattering at the sample's surface, without assuming any experimental imperfections of the "cryogenic optical setup".

For instance, they assume 100% recollection efficiency for the lasers, yet even a slight deviation from this value will completely saturate the cryostat cooling power. To clarify my statement, the light needs to be guided out of the mixing chamber: if it passes through a small view window, it will not be completely transmitted. The light scattered back will be confined in the cryogenic environment and eventually absorbed.

I still believe this is an important point, since the focus on cryogenic quantum applications, however, I cannot exclude that future improvements in optical setups will overcome such limitations.

We are pleased that we have addressed the Reviewer's remaining comments, and we completely agree with all of this last point. The SI highlights the value of our approach fundamentally over others and the presented estimates are consistent with measurements from the literature. Nonetheless, we have also now added a specific note to the revised supplementary information with the Reviewer's comment above, that the best case estimate can only be achieved with an optical setup that is as sufficiently advanced as described.